# Avian Pattern Recognition Receptor Sensing and Signaling

**DOI:** 10.3390/vetsci7010014

**Published:** 2020-01-27

**Authors:** Sabari Nath Neerukonda, Upendra Katneni

**Affiliations:** 1Department of Animal and Food Sciences, University of Delaware, Newark, DE 19716, USA; upendra@udel.edu; 2Current Address: Center for Biologics Evaluation and Research, U.S. Food and Drug Administration, Silver Spring, MD 20993, USA

**Keywords:** innate immunity, pattern recognition receptors, pathogen sensing, signaling

## Abstract

Pattern recognition receptors (PRRs) are a class of immune sensors that play a critical role in detecting and responding to several conserved patterns of microorganisms. As such, they play a major role in the maintenance of immune homeostasis and anti-microbial defense. Fundamental knowledge pertaining to the discovery of PRR functions and their ligands continue to advance the understanding of immune system and disease resistance, which led to the rational design and/or application of various PRR ligands as vaccine adjuvants. In addition, the conserved nature of many PRRs throughout the animal kingdom has enabled the utilization of the comparative genomics approach in PRR identification and the study of evolution, structural features, and functions in many animal species including avian. In the present review, we focused on PRR sensing and signaling functions in the avian species, domestic chicken, mallard, and domestic goose. In addition to summarizing recent advances in the understanding of avian PRR functions, the present review utilized a comparative biology approach to identify additional PRRs, whose functions have been well studied in mammalians but await functional characterization in avian.

## 1. Introduction

The innate immune system acts as the first line of defense against invading microorganisms and plays an important role in the establishment of anti-microbial state or shaping the adaptive immune responses. During the evolutionary arms race of hosts and their microorganisms, hosts have evolved receptors that can sense and respond to conserved endogenous or exogenous patterns. Pattern recognition receptors (PRRs) are germline-encoded receptors that sense conserved pathogen-associated molecular patterns (PAMPs) shared among many microorganisms or endogenous damage-associated molecular patterns (DAMPs) to initiate downstream signaling. The current list of PRRs and their functions is extensive, with novel PRRs and PRR functions unraveled periodically. PRRs are notably classified into four families: Toll-like receptors (TLRs), nucleotide oligomerization domain (NOD)-like receptors (NLRs), C-type lectin receptors (CLRs), and retinoic acid-inducible gene I (RIG-I)-like receptors (RLRs). PRR ligation of PAMPs or DAMPs triggers multiple signaling pathways that culminate in the activation of nuclear factor-κB (NFκB), mitogen-activated protein kinases (MAPKs), and interferon regulatory factors (IRFs), which facilitate expression of pro-inflammatory cytokines and chemokines, type I interferons (IFNs) and other molecules necessary for shaping adaptive immune responses. A robust anti-microbial environment is established in response to pro-inflammatory cytokines and type I IFNs.

With the advent of whole genome sequencing, comparative genome analysis, RNA-sequencing profiling, and canonical PRR ligand induced responses, many studies have delineated PRR diversity and functions in avian species of economic importance including domestic chicken, mallard, and goose. Mammalian and avian genomes share several evolutionary conserved syntenic regions containing PRRs, as they diverged approximately 300–400 million years ago. In addition, avian lineages also comprise PRRs with unique molecular and functional properties that enable distinct and species-specific functions. In the present review, we tended to focus on these properties of avian PRRs including TLRs, NLRs, and RLRs whose functions are characterized or yet to be defined. For those avian PRRs that await functional characterization, we extrapolated PRR functions on the basis of sequence alignments that led us to identify conserved or shared domains, motifs, and functional residues with mammalian counterparts. Previous reviews on avian PRR functions were limited in their scope, and they focused on specific PRRs (TLRs) in a single species (chicken) [1] or functions of a subset of PRRs in chicken, duck and goose [2], and many reviews focused on the role of PRRs in the context of different viral infections [3,4,5,6,7]. The present review aims to update major advancements since the latest reviews in 2013, which included PRRs and downstream adaptors identified in duck and goose as a result of whole genome sequencing of these species. In addition, the present review comprehensively spans TLRs, RLRs, NLRs, and additional DNA or RNA sensors in chicken, duck, and goose, and highlights areas for further research that may enhance the understanding of PRR functions and evolution (Table 1, Figure 1). For age-dependent and cell type-specific differences in PRR expression, the readers may refer to the recent review in [8]. In the sections described herein, we first introduce mammalian PRR sensing and signaling mechanisms that are well studied. We next describe functional similarities or differences of avian PRRs in the order of *Gallus gallus domesticus* (domestic chicken), *Anas platyrhynchos* (mallard duck), and *Anser cygnoides domesticus* (domestic goose).

## 2. Toll-Like Receptors

### 2.1. General Principles of Toll-Like Receptor Sensing and Activation

TLRs are evolutionarily conserved PRRs and are present in all living multicellular organisms. A total of 10 (TLR1–TLR10) and 12 (TLR1–TLR9, TLR11–TLR13) TLRs have been identified in human and mouse, respectively. All identified TLRs are type I transmembrane (TM) proteins with a conserved domain organization [9,10]. Each TLR is composed of an N-terminal ectodomain with leucine-rich repeats (LRRs) that mediate PAMP or DAMP recognition, a TM domain that positions it in membrane, and a cytoplasmic Toll/interleukin-1 receptor (TIR) domain that binds to TIR domain-bearing adaptors to mediate downstream signaling [11]. The number and glycosylation patterns of LRRs (19-27) can vary, and variations in LRRs create binding pockets or regions with variable PAMP ligand specificity [9]. Most mammalian TLRs function as homodimers, although TLR2 heterodimerizes with either TLR1 or TLR6 [12]. Mammalian TLR4 functions in complex with MD-2 co-receptor and signals effectively in the presence of CD14 lipid scavenger protein, lipopolysaccharide (LPS) ligand, and LPS binding protein (LBP) that enhance ligand affinity towards CD14 [13]. Similarly, several other TLRs signal most effectively in the presence of CD14, including TLR7 and TLR9 [14]. 

TLRs are localized on the cell surface or in intracellular compartments such as the endoplasmic reticulum (ER), endosome, lysosome, or endolysosome, where they recognize distinct or overlapping PAMPs and DAMPs such as glycolipid, lipoprotein, protein, and nucleic acid components [15]. TLR localization on cell surface and in intracellular compartments is critical for ligand recognition and preventing recognition of nucleic acids of self, which can contribute to autoimmunity or aberrant inflammation [16]. TLR localization is determined by the presence of targeting motifs in the linker region between the TM and TIR domain (TLR3), TM domain (TLR7), and cytoplasmic domain (TLR9) [17,18,19]. In addition, TLR localization is also controlled by proteins such as UNC93B (TLR9), MD-2, gp96 chaperone, and PRAT4A (TLR1, TLR2, TLR4, TLR7, and TLR9) [17,18,19]. Finally, endosomal TLRs undergo proteolytic cleavage by cathepsins B, S, L, H, and K and asparginyl endopeptidase to attain a mature functional form that mediates ligand recognition and downstream signaling [20,21,22]. In this regard, following cleavage of TLR9, N-terminal-cleaved fragment of ectodomain (TLR9N) remains associated with truncated TLR9 (TLR9C) to form a complex, which acts as a bona fide DNA sensor [22].

TLR ligands are mostly of microbial origin that are critical for the survival of microbe and absent in the host. This feature allows detection of a wide range of conserved microbial patterns belonging to bacteria, viruses, protozoa, and fungi. TLR ligands include lipid conjugated sugars (glycolipids) or proteins (lipopeptides) recognized by TLR4/MD-2 complex (LPS), and heterodimers of TLR2/TLR1 or TLR2/TLR6 (di- and tri-acylated lipopeptides, respectively) [15,23]. Similarly, bacterial flagellin is recognized by human TLR5 and mouse TLR11, respectively [24,25,26]. Lastly, host or microbial nucleic acids are recognized by TLR3 (double-stranded RNA, dsRNA), TLR7 and TLR8 (single-stranded RNA, ssRNA), TLR9 (DNA), and mouse TLR13 (23S ribosomal RNA) [27,28,29,30].

Upon TLR ligand engagement, individual TLRs differentially recruit specific members of TIR domain-containing adaptors such as MyD88, TRIF, TIRAP/MAL, or TRAM. MyD88 functions as an adaptor for all TLRs and activates downstream NF-kB and MAPKs for pro-inflammatory cytokine induction. TIRAP is a sorting adaptor that facilitates recruitment of MyD88 to cell surface TLRs (TLR2 and TLR4) or endosomal TLR9 through its lipid-binding domain that promotes binding to phosphatidylinositol 4,5-bisphosphate (PI(4,5)P2) at the plasma membrane or PI(3)P on endosomes, respectively [31,32]. By binding to different lipids at plasma membrane and endosome, TIRAP facilitates formation of functional TLR4 and TLR9 signaling complexes at the respective sites [31].

On the other hand, TRIF is selectively recruited to TLR3 and TLR4, leading to selective activation of IRF3, NF-kB, and MAPKs for induction of type I IFNs and proinflammatory cytokines [33]. TRAM is selectively recruited to TLR4 to link TLR4 and TRIF, whereas TLR3 directly interacts with TRIF upon phosphorylation of the two tyrosine residues in the cytoplasmic domain of TLR3 by the ErbB1 and Btk kinases [34,35,36]. Collectively, TLR signaling can be either Myd88-dependent or TRIF-dependent.

### 2.2. Avian TLRs

#### 2.2.1. TLR1/2

Previous comparative analysis of chicken TLR1 type 1 (chTLR1t1) and the known human TLR (hTLR) receptors at the protein level revealed that the chTLR1t1 protein closely resembled hTLR1, hTLR6, and hTLR10 with amino acid similarity values of 83% for the predicted cytoplasmic TIR and TM domains of chTLR1t1 and 60% for the extracellular domain [37,38]. Comparative analysis of predicted TLR1t1 homologs of duck (duTLR1t1) and goose (goTLR1t1) with known hTLR1/6/10 revealed lower amino acid similarity values of 74.2% and 75.3% for the predicted TIR domain, and 63.7% and 62.5% for the predicted TM domain, respectively. A truncated form of TLR1t1, named TLR1t2, was identified in chicken and duck but not goose [37].

Earlier studies defining hTLR2 and hTLR6 functional interactions constructed dominant negative mutants of hTLR2/6 by mutating proline at position 688 to histidine [39]. Furthermore, previous studies in human defined the critical requirement of glycine residue at position 689 in BB loop of hTLR1 TIR domain to facilitate interaction with 748RF749 residues of DD loop of hTLR2 TIR domain, which contributed to TLR signaling [40]. Functional importance of proline and glycine at 688 and 689 positions, respectively, is highlighted by their conservation in chicken, duck, and goose TLR1t1 [37]. In addition, the juxtamembrane cysteine-rich (LRR-CT) region implicated in the regulation of TLR signaling and 19 consecutive LRR/LRR-like motifs observed for hTLR1, hTLR6, hTLR10, and chTLR1t1 remained conserved in duck and goose as well. Finally, a 12-amino acid stretch located in LRR10 of chTLR1t1 that is absent in hTLR1/6/10 is also located in LRR10 of duck and goose, indicative of similar or shared ligand specificity between avian TLR1t1 homologs (Appendix A) [41]. Finally, prediction of putative *N*-glycosylation sites using the NetNGlyc 1.0 (www.cbs.dtu.dk/services/NetNGlyc) and ScanProsite (expasy.org/tools/scanprosite) algorithms revealed seven and eight asparagine residues in the duTLR1t1 and goTLR1t1 ectodomain, respectively, compared to five residues in chTLR1t1 indicative of variable degree of glycosylation.

Functional characterization of chTLR2 and chTLR1 proteins expressed individually or in combination indicated that they form functional heterodimers but not homodimers [37]. For instance, chTLR2t1 can form a functional heterodimer with chTLR1t2 or chTLR1t1 and, similarly, chTLR2t2 can form a functional heterodimer with chTLR1t1 or chTLR1t2. According to one study, although only chTLR2t1/chTLR1t2 complex efficiently responded to peptidoglycan (PGN), unlike the mammalian TLR2/TLR1 or TLR2/TLR6 complexes, chicken TLR1/2 combinations (chTLR2t1/chTLR1t2, chTLR2t2/chTLR1t1, and chTLR2t2/chTLR1t2) efficiently responded to both di- and tri-acylated lipopeptides and sonicated *Mycobacterium avium* extracts [37]. In this regard, transfer of LRR6-16 of chTLR1t1 to hTLR1 was sufficient to confer broad ligand specificity to hTLR1/chTLR2t2 complex, indicative of critical importance of this region in determining ligand specificity [37]. In addition, previous findings noted that chTLR2t2 but not chTLR2t1 formed a functional complex with chTLR1t1 due to strict structural constraints to facilitate interactions. The chTLR2t1 and chTLR2t2 differ mainly in a 200-aa stretch spanning LRR8–14, suggesting critical involvement of this region in mediating functional interactions. Finally, a several-fold enhancement of chTLR2t2/chTLR1t1 response to di- and tri-acylated lipopeptides was noted in the presence of mammalian lipid scavenger CD14 [42]. Whether chicken CD14 contributes to similar enhancement is currently unknown and is pending investigation.

The reasons for the evolution and preservation of two isoforms of TLR2 and TLR1-like proteins in chicken or duck remain unknown. Although chTLR2/chTLR1 complexes clearly exhibit broad ligand specificity compared to their mammalian counterparts, ligand-induced responses of different receptor combinations were quantitatively different. Quantitative differences in ligand responsiveness and tissue specific expression of various receptor subunits clearly indicate distinct functions of these complexes [37,43]. Finally, amino acid polymorphisms in chTLR1t1 and chTLR1t2 proteins were identified in various breeds of chickens, although the implications of this diversity for disease resistance remains obscure [44].

Although ducks evolved and maintained two isoforms of TLR2 that arose upon gene duplication, sequence information available for goose points to a single TLR2 gene isoform similar to goose TLR1 [2,45]. Functional characterization of duck and goose TLR1/2 combinations and ligand specificity is pending investigation.

#### 2.2.2. TLR3

Mammalian TLR3 is an endosomal TLR that senses viral dsRNA or dsRNA analog (poly I:C) to promote homodimerization via lateral surface interactions. TLR3 stimulation by dsRNA results in downstream signaling in a TRIF-dependent manner to result in type I IFN release [46]. Released type I IFNs function to induce antiviral state in virus-infected cells and IFN-exposed uninfected cells by stimulating induction of hundreds of interferon-stimulated genes (ISGs) that inhibit multiple steps of viral replication beginning particle entry until release [47].

The genomes of chicken, duck, and goose contain a single ortholog of mammalian TLR3. Comparison of the genomic sequences of human and chicken led to the identification of a single chicken TLR3 (chTLR3) protein that is 61% identical to its human ortholog [48,49,50]. Preliminary studies indicated that chicken splenocytes and leucocytes respond to poly I:C, resulting in the production of type I IFNs that diminished upon small interfering RNA (siRNA)-mediated knockdown of chTLR3 [48,49,50]. Furthermore, ectopic expression of chTLR3 conferred responsiveness to HEK293 cell line, resulting in activation of NFκB [49]. Similar to mammals, chTLR3 is an ISG and pretreatment with IFN-α-sensitized DF-1 cells to induce an enhanced type I IFN response upon poly I:C stimulation [50]. In addition, although not all the components of mammalian TLR3 signaling cascade have been identified in chicken genome, orthologs of crucial components such as TRIF and TBK1/IKKε have been found, indicative of grossly identical signaling mechanisms in chicken [51,52]. Finally, 30 gene variants of chTLR3 were found, of which 16 had non-synonymous substitutions. Amino acid substitutions such as S180G and K240T were predicted to form *N*-myristoylation site and loss of one protein kinase C phosphorylation site, respectively, although functional relevance of these substitutions in disease resistance or susceptibility is still unknown [53].

Peking duck TLR3 (duTLR3) has a high amino acid sequence similarity with the TLR3 protein sequences of Jing ding duck (99.6 %), Muscovy duck (97.1 %), and chicken (86.3 %), whereas moderate similarity with the human and mouse TLR3 proteins (62.3 and 60.0 %, respectively) was found [54,55]. duTLR3 is a typical type I TM protein and exhibits a similar domain structure to TLR3 of other species with an extracellular LRR domain at N-terminus, a TM domain, and an intracellular TIR domain at the C-terminus. Although upregulation of duTLR3, NFκB, IFNα, and ISGs in spleens of Peking ducklings and upregulation of duTLR3 in different tissues of highly pathogenic avian influenza (HPAI)-infected A/Duck/Guangdong/212/2004 (H5N1) Muscovy ducks has been demonstrated, functional studies with TLR3 ligands in duck cells are still awaited [55].

Goose TLR3 (goTLR3) is a type I TM protein that is 896 amino acids in length and consists of a N-terminal signal peptide, 14 leucine-rich repeat (LRR) domains, a leucine-rich repeat C-terminal (LRR-CT) domain, a TM domain, and a 147-amino acid TIR domain in its carboxy-terminus. goTLR3 shared 84.4%, 92%, 59.1%, and 56.9% similarity with chicken, duck, human, and mouse TLR3, respectively. Like mammalian and chTLR3, transfection of goTLR3 into HEK293 cells and poly I:C stimulation conferred responsiveness and NFκB activation indicative of similar ligand specificity [56]. Finally, TLR3 was upregulated in various tissues of Magang geese upon HPAI-infection [56].

#### 2.2.3. TLR4

Mammalian TLR4 is part of a trimolecular LPS receptor complex that consists of two coreceptors: MD-2, that is essential for TLR4 function, and CD14, which is a high-affinity glycosyl-phosphatidyl inositol-linked ligand-binding protein. The primary ligand of mammalian TLR4/MD-2 receptor complex is a major Gram-negative bacterial surface component, LPS, which primarily binds to serum LPS-binding protein (LBP) and is then transferred to TLR4/MD-2 complex by CD14 [57,58]. With regards to ligand specificity, human TLR4/MD-2 specifically recognizes penta-acylated LPS forms, whereas murine TLR4/MD-2 recognizes both penta- and hexa-acylated LPS forms [59]. This differential ligand specificity is due to the central hypervariable region in the extracellular domain of human TLR4/MD-2 complex. In contrast to human TLR4/MD-2, murine TLR4/MD2 recognizes lipid_IVA_ and plant-derived taxol in a MD2-dependent manner. LPS binding causes receptor dimerization and activation leading to downstream signaling. Activation of mammalian TLR4/MD-2 receptor complex leads to two different downstream signaling pathways, the MyD88/TIRAP pathway and the TRAM/TRIF pathway [33,36]. Activation of the MyD88/TIRAP pathway leads to NFκB activation and immediate-early production of pro-inflammatory cytokines, whereas activation of the TRAM/TRIF pathway results in delayed NFκB activation and production of type I IFNs through activation of the IRF3, which leads to endotoxic shock [33,36].

Cloning and characterization of chicken TLR4 (chTLR4)/MD2 complex indicated cell surface expression and ligand specificity largely similar to murine TLR4/MD2 rather than human [1]. chTLR4/MD2 responded to LPS components from various bacteria in addition to synthetic lipidIVA ligands [1]. The differential recognition of lipid IVa (compound 406) by human and murine MD-2 was attributed to amino acids 57, 61, and 122 of murine MD-2 [60]. Although it is highly likely that LPS specificity is determined by TLR4/MD2 complex rather than just MD2, only two of the amino acid residues (57 and 122) are conserved in chicken MD2, whereas amino acid 61 is different from both human and murine MD-2 [1]. Another peculiar feature of chTLR4/MD2 activation is the lack of TRAM in chicken, and thus absence of TRAM/TRIF signaling pathway and type I IFN production [1]. This feature was proposed as contributing to the general resistance of chicken to endotoxic shock [38]. However, studies in peripheral blood mononuclear cells (PBMCs) derived from Aseel and Ghagus Indian breeds of chicken revealed enhanced upregulation of TRIF and TRIF-dependent genes (*TRIF, IRF7, and IFNβ*) upon LPS treatment, indicative of existence of TRIF signaling pathway that is MyD88-independent [61]. Finally, cloning and characterization of chicken CD14 indicated it to be a TM protein unlike glycosylphosphatidylinositol (GPI)-anchored human CD14 [62]. Chicken CD14, however, shared other features of mammalian CD14, such as 8 conserved cysteines with the potential to form 4 disulphide bridges, and 11 conserved LRRs with an additional chicken-specific LRR [62]. These features of chicken CD14 point to a compromised coreceptor function or an independent receptor function [62].

Duck TLR4 (duTLR4) gene encodes an 833-amino acid protein that displayed identities of 82.1% with chTLR4, 96.8% with goose TLR4 (goTLR4), and 43.2–45.2% with mammalian homologs. Although duTLR4’s sub-cellular localization, species-specific differences, and its response to various LPS or lipid A ligands remains to be investigated, LPS treatment of duck embryo fibroblasts was found to lead to transcriptional induction of pro-inflammatory cytokines IL-1β and IL-6 [63,64]. Finally, motif prediction analysis indicated typical TLR4 domain structure with 8 LRRs and LRR-CT in the extracellular region, along with TM and cytoplasmic TIR domains [63,64].

goTLR4 gene encodes an 844-amino acid protein that shared domain structure with chicken and duck with 8 LRRs, LRR-CT, TM, and cytoplasmic TIR domains [64]. Functional CD14 orthologs in duck and goose are yet to be identified or likely absent.

#### 2.2.4. TLR5

Chicken TLR5 (chTLR5) exhibits 69% and 68% overall similarity to human and murine TLR5, respectively, with greatest similarity (72%) for the TM and cytoplasmic tail (TIR) regions [65]. chTLR5 contains 22 consecutive LRRs including LRR-CT with conserved four spaced cysteine residues that are essential for TLR5 signaling. Interestingly, chTLR5, similar to duck and goose, harbors LLRs that differ from the prevailing LRR motif (XLXXLXLXXNXφXXφXXXXFXXLX) [65]. The three LRRs that differ from the canonical motifs include LRR2 (135S, 137L), LRR3 (162Q), and LRR10 (551L) with respective amino acid differences. The extracellular domain of chTLR5 was predicted to contain eight potential *N*-glycosylation sites compared to six and nine in human and murine TLR5, respectively [65]. A highly conserved proline (position 737) in the cytoplasmic TIR domain BB-loop involved in binding MyD88 adaptor protein is also conserved in chicken, duck, and goose TLR5, suggestive of MyD88 adaptor utilization by TLR5 in these species [65]. chTLR5 transfection into HeLa 57A cell line and live *Salmonella enteritidis* SE 706 infection resulted in a dose-dependent activation of NFκB reporter. ChTLR5 exhibited superior ligand responsiveness like mouse TLR5 upon treatment with recombinant flagellin of *Salmonella typhimurium* compared to human TLR5 [65]. In contrast, this species-specific difference was not observed for recombinant flagellin of *S. enteritidis*, indicative of flagellin-specific responses. Mutational scanning of *S. typhimurium* flagellin protein revealed that although TLR5 of various host species recognize the same alpha-helical region, subtle differences in flagellin protein may contribute to differential innate immune responses in the chicken and humans [65,66,67]. Lending further support, chTLR5 was demonstrated to have relaxed ligand specificity compared to human or mouse TLR5. For instance, chTLR5 detects and responds to recombinant flagellins that contained N- (ND1a and ND1b) and C-terminal (CD1) helices with either *S. typhimurium* or *Campylobacter jejuni* β-hairpin [66]. Ligand specificity of human and mice TLR5, on the other hand, was determined by the β-hairpin structure differences, which contribute to human (or mouse) TLR5 immune evasion of *C. jejuni* flagellin [67].

Duck and goose TLR5 genes were cloned and functionally characterized to show that they exhibit greater amino acid identity to chTLR5 (87.4% and 82.7%, respectively) rather than human (50.5% and 49.8%, respectively) and mouse (50.6% and 49.9%, respectively) [68,69,70]. Furthermore, transfection of individual TLR5 genes into HeLa and HEK293 cells conferred responsiveness to *S. typhimurium* flagellin treatment [68,70].

#### 2.2.5. TLR7/8

Chicken TLR7/8 gene locus is syntenic with mammalian TLR7/8 locus with the exception that only TLR7 gene encodes a functional TLR7 ortholog, whereas TLR8-like sequences are disrupted by a 6.1 kb insertion bearing homology to a viral reverse transcriptase gene from the chicken repeat 1 (CR1) mobile element and contained stop codons [71]. Chicken TLR7 (chTLR7) displayed 62% overall amino acid identity and similar domain organization to human TLR7, with a conserved proline at 712 position, and 72% identity within the TIR domain [71]. Although functional characterization of chTLR7 is pending investigation, chicken splenocytes, PBMC, and HD11 macrophages were demonstrated to respond to TLR7 ligands, including R848, loxoribine, and poly (U) to result in primarily IL-1β and IL-6 pro-inflammatory cytokine production and less type I IFNs [72,73,74]. TLR7 ligand responsiveness was sensitive to chloroquine treatment, indicative of similar endosomal localization of chTLR7 as mammalian TLR7 [71]. Furthermore, the inhibitory effects of TLR7 ligand-induced antiviral state on replication of influenza and New Castle’s disease virus was also demonstrated [71].

Duck TLR7 gene was cloned and characterized from White Peking ducks. Duck TLR7 (duTLR7) displayed 44% and 85% amino acid identity to human and chTLR7, respectively, with largely identical domain organization [75]. Conserved domain features include ectodomain composed of tandem LRRs, a TM domain, and an intracellular TIR signaling domain. In addition, duTLR7 contains variants of conserved boxes within the TIR region critical for TLR7 signaling (box 1 and 2) FDAFISY and GYKCC-RD-PG, and localization (box 3)—a W flanked by basic residues. Treatment of duck splenocytes with TLR7 ligands loxoribine and R848 resulted in the production of IL-1β, IL-6, and type I IFNs, suggesting that duTLR7 is a true ortholog of mammalian TLR7, although further functional characterization in mammalian cells and in duck cells via ectopic expression and gene disruption remains to be carried out [75]. Interestingly, downstream of TLR7 gene, fragmented sequences of TLR8 were identified by BLASTX search [75]. Upon search for repeat sequences using Repeat Master analyses, this region was found to contain several retroviral insertion elements, including a 1.4 kb CR1 element present between fragments of the TLR8 gene [75]. Duck TLR8 gene is thus non-functional.

Goose TLR7 (goTLR7) displayed greater than 80% identity to chicken and duck and least identity (60%) to human TLR7 and an overall conserved domain structure [73,74]. Goose spleen mononuclear cells treated with TLR7 ligands R848 and imiquimod exhibited significant induction of IL-1β, IL-6, and IFN-α, and infection of Sichuan white geese with new gosling viral duck enteritis virus (NGVEV) led to a significant induction of TLR7 expression in various tissues including thymus, bursa, spleen, peripheral blood leukocytes, and small intestine [73]. Further studies must focus on functional characterization of goTLR7, its ligand responsiveness, and downstream signaling in HEK293 or HeLa cells. Finally, a preliminary BLASTX search of 20 kb sequences downstream of goTLR7 revealed TLR8 gene fragments and sequences bearing homology to reverse transcriptase and RNA-dependent DNA polymerase proteins that remain part of retroviral insertion elements, suggestive of potential TLR8 gene fragmenting and disruption in goose, as observed in chicken and duck. TLR8-CR1 insertion elements are identified in galliform birds but not non-galliform birds [71]. Since the ancestor of galliform birds diverged from non-galliform birds approximately 90 million years ago, it is evident that a major avian vertebrate lineage has lost functional TLR8 gene [76].

#### 2.2.6. TLR15

TLR15 is an evolutionarily old receptor that was first identified in chicken and then in remaining avian and most reptilian species, including lepidosaurians and archosaurians [77,78]. On the basis of the identification of putative TLR15 ortholog in Australian ghost shark species, it is hypothesized that TLR15 did not arise in the sauropsid lineage but rather originated before the divergence of Chondrichthyes fish and tetrapods. Furthermore, the large-scale gene loss of TLR15 from the teleost fish, amphibian, and mammalian lineages coincides with the variable leucine codon usage among avian and reptilian TLR15s or identified TLR15-like remnant sequences in turtles, which are genetically closely related to crocodiles and birds [77,79]. Avian and reptilian TLR15 share a conserved feature of activation by proteases resulting in MyD88-dependent downstream activation of NFκB. Although avian TLR15 is expressed on the cell surface, reptilian TLR15 displays intracellular localization [77].

Chicken TLR15 (chTLR15) has typical TLR architecture with a signal sequence, an N-terminal LRR domain, a TM helix, and a TIR domain. Furthermore, TLR15 TIR domain contains a conserved BB-loop structure, including a conserved proline residue known to be essential for MyD88-dependent signaling in mammalian TLRs (Appendix A) [80]. Cloning, characterization, and ligand screening revealed that chTLR15 is activated upon sensing proteolytic activity at the cell surface in human or COS7 cells. Both bacterial and fungal proteases were demonstrated as activating TLR15, resulting in NFκB activation and cytokine production in a MyD88-dependent manner. Protease-induced activation of chTLR15 resulted in its proteolytic cleavage and appearance of a 70 kDa C-terminal chTLR15-cleaved fragment that is sensitive to PMSF serine protease inhibitor treatment [80]. Current model of chTLR15 ligand sensing and activation suggests that protease sensing and cleavage of extracellular LRR domain results in the release of inhibitory elements to cause chTLR15 self-activation without the ligand requirement. In line with chTLR15 activation by bacterial proteases, previous studies demonstrated significant upregulation of chTLR15 in the cecum of *Salmonella enterica serovar Typhimurium*-infected chickens and in chicken embryo fibroblasts stimulated with heat-killed *S. enterica serovar Typhimurium* [81,82]. Interestingly, one study demonstrated specific upregulation of chTLR15, inducible nitric oxide synthase (iNOS), and nitric oxide production in HD11 macrophages and primary chicken chondrocytes upon *Mycoplasma synoviae* infection or treatment with its diacylated lipopeptide [83]. These effects were sensitive to siRNA-mediated knockdown of TLR15 [83]. Although chTLR15 is known to form homodimers in the context of protease-mediated activation, whether it heterodimerizes with other TLRs in sensing other ligands remains to be investigated.

Like chicken, duck and goose TLR15 displays gene synteny and flanking by *psme4, erlec1, gpr75, chac2,* and *asb3* genes [79]. Upon alignment with chicken TLR15, duck and goose TLR15 displays identical domain structure, including signal sequence, an N-terminal LRR domain, 19 LRRs, LRR-CT, a TM helix, a TIR domain, and conserved proline box (Appendix A). One peculiar feature of duck and goose TLR15 includes the deletion stretches of 9 and 5 amino acids in the LRR3 region (Appendix A). Functional characterization and ligand specificity of duck and goose TLR15 awaits investigation.

#### 2.2.7. TLR21

Avian TLR21 is a functional ortholog of mammalian TLR9 with very little similarity at the amino acid level. Chicken TLR21 however shares amino acid similarity with TLR21 of amphibians (*Xenopus tropicalis*—61% identity) and fish (*Takifugu rubripes*—57% identity) [84]. Like TLR9, chTLR21 expression in HEK293 cells resulted in its ER localization and activation of NFκB in response to synthetic CpG-ODN treatment. Despite low amino acid similarity with mammalian TLR9, TLR21 also functions as an unmethylated CpG DNA sensor, suggesting convergent evolution of different TLR proteins. Compared to human and mouse TLR9, chTLR21 exhibits broad ligand specificity and recognizes a wide repertoire of synthetic CpG DNA molecules, including the ones bearing GTCGTT motif (CpG ODN 2006) and the GACGTT motif (CpG ODN 1826) and even responds to bacterial chromosomal DNA. Furthermore, ligand responsiveness of chTLR21 in HD11 macrophages is sensitive to chloroquine treatment pointing to ER and endolysosomal functions. In human and mouse, UNC93B1 was demonstrated to control the stability and endosomal localization of TLR3, TLR7, and TLR9 [85]. Although human and mouse lack TLR21, zebra fish genomes contain an ortholog of mammalian TLR9, as well as fish-specific TLRs including TLR21, whose localization and activation was found to be regulated by UNC93B1 [85]. Whether avian TLR21 shares similar conserved mechanism involving UNC93B1 to control the localization and activation of TLR21 is still unknown. Finally, whether chTLR21 requires proteolytic cleavage for activation, as described for human TLR9, remains to be determined [38].

Cloning and characterization of Cherry valley duck TLR21 (duTLR21) indicated that it shares 92% identity with goose TLR21 (goTLR21) and 76% identity with chTLR21 with 20 LRR motifs; a signal peptide; a TM domain; and an intracellular TIR domain comprising box 1, box 2, and box 3, three conserved motifs [86]. Like chTLR21, duTLR21 exhibited broader ligand specificity and responded to CpG-ODN through NFκB activation and downstream transcription of IL-1β, IL-6, and IFN-α [86]. Lastly, ectopic expression of duTLR21 in DEF cells conferred an antiviral effect against duck plague virus (DPV) infection [86].

Like chTLR21 and duTLR21, goTLR21 is a CpG DNA sensor. Treatment of goose PBMCs with CpG ODN2006 resulted in a significant upregulation of proinflammatory cytokines (IL-6 and IL-1β) and interferons (IFN-α and IFN-γ) [87]. Unlike low pathogenic avian influenza (LPAI)-infection, infection of goose PBMCs and goslings with NGVEV resulted in a significant upregulation of goTLR21 [87].

## 3. NOD Leucine-Rich-Repeat Containing Protein Receptors (NLRs)

### 3.1. General Principles of NLR Activation and Inflammasome Assembly

Apart from TLRs, mammalians encode PRRs, nucleotide-binding oligomerization domain (NOD) leucine-rich-repeat containing protein family receptors (NLRs) that activate multiprotein signaling complexes known as inflammasomes upon sensing PAMPs, DAMPs, and cellular cues indicative of altered homeostasis [88]. Like NLR family proteins, AIM2 and pyrin also function as PRRs that facilitate inflammasome activation and assembly. Non-mammals including avian and monotremes lack AIM2, whereas no true pyrin ortholog was identified in avian thus far [89]. Canonical inflammasome assembly and activation is generally dependent on two functional units. The first unit involves an adaptor protein such as ASC that facilitates NLR oligomerization, and the second unit is an effector enzyme that catalyzes downstream inflammatory responses [90]. A significant number of inflammasomes including NLRP1, NLRP3, NLRP6, AIM2, and pyrin utilize ASC as an adaptor through pyrin-pyrin interactions between PRR and adaptor to result in their oligomerization [88]. The caspase activation and recruitment domain (CARD) of ASC then engages the CARD domain of effector enzyme pro-caspase-1 to recruit it into the oligomeric complexes and induce proximity-induced self-cleavage of pro-form of caspase 1 into active form. On the other hand, PRRs of NAIP family also assemble into inflammasomes by engaging NLRC4 adaptor through NACHT domain interactions between PRR and adaptor to result in oligomerization, as well as recruitment and activation of caspase 1 [88,91]. Activated caspase 1 cleaves pro-forms of IL-1β and IL-18 into active forms, which are released into the extracellular space by means of membrane pores formed by caspase 1-cleaved protein known as gasdermin D (GSDMD) [91]. Finally, NLRs such as NLRX1, NLRC3, and NLRC5 have functions distinct from inflammasome assembly, for instance, negative regulation of innate signaling and MHC I transactivation, respectively [92,93,94]. Currently available genome sequences reveal that avian encodes functional homologs of NLRX1, NLRP3, NLRC3, and NLRC5, whose features are addressed below.

### 3.2. Avian NLRs

#### 3.2.1. NLRX1

Due to the lack of characterization of N-terminus, the “X” nomenclature was used to define the mammalian NLRX1 N-terminus [88]. The N-terminus of NLRX1 was found to harbor a mitochondria-targeting sequence (MTS) [95]. Following the “X” domain is a NACHT domain that facilitates oligomerization. The C-terminus of NLRX1 is unique in that it consists of seven LRRs followed by an uncharacterized three-helix bundle. Mammalian NLRX1 is a multifunctional protein that was found to modulate a multitude of innate signaling and autophagy-related pathways. In mitochondria, NLRX1 was found to interact with the complex III-associated protein UQCRC2 to facilitate reactive oxygen species (ROS) production. ROS in turn activate the JNK pathway to promote apoptosis [92]. During viral infection, NLRX1 was also demonstrated to negatively regulate RIG-I activation through interaction with poly(rC) binding protein 2 (PCBP2) and mitochondrial anti-viral signaling (MAVS) via its NACHT domain to cause lysine 48 (K48)-linked polyubiquitination and degradation of MAVS [92]. This ultimately resulted in attenuated production of IL-6 and type I IFN. In addition, upon association with MAVS and TUFM, NLRX1 promotes autophagy and augmented negative regulation of type I IFN production [96]. NLRX1 also inhibits NFκB signaling by interacting with IkB kinase (IKK) and by promoting TRAFasome formation. NLRX1 may also negatively regulate the MAPK pathway. Furthermore, recently, NLRX1 was demonstrated to promote *Listeria monocytogenes*-induced mitophagy by interacting with LC3 via LC3 interacting region motif (LIR: EEFQLL) to promote bacterial survival [97]. Conversely, NLRX1 deficiency enhanced the number of damaged mitochondria, enhanced mitochondrial production of reactive oxygen species (ROS), and reduced bacteria survival. Lastly, in the presence of TNF, NLRX1 interacts with caspase-8 to induce TNF-induced apoptosis, and this interaction may inhibit complex I and III of the electron transport chain.

NLRX1 proteins of chicken, duck, and goose shared similar domain architecture with human in that they contained an N-terminal MTS (as determined by MitoProt program), NACHT oligomerization domain and LIR, and a C-terminal LRR (Appendix A). In addition, the C-terminal region arginine of NLRX1 implicated in RNA binding (Arg699) was conserved in avian species suggestive of similar interactions with poly(I:C) and ssRNA [98]. Functional characterization of avian NLRX1, its ligand responsiveness, and immune regulatory mechanisms awaits investigation.

#### 3.2.2. NLRP3

Mammalian NLRP3 was found to respond to a diverse set of crystalline and particulate ligands including uric acid crystals, silica, asbestos, alum, extracellular ATP, bacterial pore-forming toxins, RNA/DNA hybrids, mitochondrial DNA (mtDNA), ROS, cytosolic potassium efflux, and pathogenic infections [99]. Further regulation of NLRP3 activation includes NEK7 binding to NLRP3, resulting in caspase 1 activation [100].

Multiple sequence alignment of avian NLRP3 with human NLRP3 revealed the presence of three characteristic domains of NLRs, including an N-terminal PYD domain, a central NACHT domain, and six C-terminal LRR domains. In the regions other than the well-characterized domains, avian NLRP3 displayed major deletions in the C-terminal LRR region (Appendix A). Among avian species, cherry valley duck NLRP3 was cloned and functionally characterized, where overexpression of duck NLRP3 in DEF cells had an anti-bacterial effect towards *Escherichia coli* infection and enhanced the upregulation of IL-1β, IL-18, and TNF-α [101]. Functional characterization of chicken and goose NLRP3 is pending investigation.

#### 3.2.3. NLRC3

Mammalian NLRC3 is another negative regulator of inflammation that is generally less well studied compared to NLRX1. Initially, NLRC3 was found to interact with TRAF6 and inhibited NFκB signaling by reducing overall K63-linked ubiquitination of TRAF6 [102]. Peritoneal macrophages from *Nlrc3*-deficient mice displayed enhanced LPS responsiveness with greater levels of TNF, IL-6, IL-1β, and IL-12 [102]. Likewise, in mouse model of septic shock, *Nlrc3-*deficient mice displayed increased disease severity and circulating levels of TNF and IL-6, indicative of the negative regulatory role of NLRC3 [102]. Secondly, upon treatment of mouse bone marrow derived macrophages (BMDMs), mouse embryonic fibroblasts (MEFs), and peritoneal macrophages with immune stimulatory DNA (ISD) and poly(dA:dT), NLRC3 mediated negative regulation of type I IFN production in a TRAF6-independent manner [93]. Consequently, ISD, poly (dA:dT), and herpes complex virus 1 (HSV-1) infection of BMDMs, MEFs, and peritoneal macrophages obtained from NLRC3-deficient mice resulted in elevated IFN and IL-6 production. Furthermore, NLRC3-deficient mice are resistant to HSV-1 infection and displayed higher levels of IFN-β, IL-6, and TNF, as well as reduced HSV-1 brain titers [93]. Mechanistically, full-length NLRC3, CARD-NBD (nucleotide-binding domain), and NBD alone are strongly associated with STING, and prevented STING trafficking to perinuclear puncta upon ISD stimulation. In addition, NLRC3 also interacted with downstream TBK1 N-terminus [93]. In parallel, a strong reduction in phosphorylated-TBK1 and -IRF3 levels has been observed [93].

Although avian NLRC3 remains to be functionally characterized, on the basis of sequence alignment and domain search, the overall architecture of avian NLRC3 is identical to human NLRC3 with an N-terminal CARD domain, NBD, and a C terminal LRR region (Appendix A).

#### 3.2.4. NLRC5

Mammalian NLRC5 contains a tripartite domain structure like the remaining NLRs with an N terminal atypical CARD, a centrally located NBD, and carboxy terminal LRRs [103]. The NBD contains a Walker A motif (aka P-loop; a nucleoside triphosphate (NTP) binding site) and a Walker B motif (an NTP hydrolysis site), which are essential for the nuclear localization and transactivating function of NLRC5 [104]. In addition, NLRC5 contains a bipartite nuclear localization signal (NLS) between the CARD and NBD, and mutations within the NLS prevent nuclear import and transactivating function of NLRC5 [104]. NLRC5 expression can be induced upon IFNγ treatment of both immune and non-immune cells in a signal transducer and activator of transcription 1 (STAT1)-dependent manner [105,106]. Upon induction, NLRC5 interacts with transcription factors bound to a conserved W/SXY motif in the MHC class I promoter to form a multiprotein complex known as NLRC5 enhanceosome to influence MHC I expression [103]. In addition, conflicting reports have indicated positive and negative roles of NLRC5 in TLR4 and RIG-I signaling [76,107].

Chicken, duck, and goose NLRC5 display similar domain organization to human with atypical N-terminal CARD, NBD bearing Walker A and B motifs, and a long C-terminal LRR domain. In addition, putative bipartite NLS (RRRR and KR) was found between CARD and NBD (Appendix A). Whether avian NLRC5 has similar or distinct localization and functions to mammalian remains unknown and is a subject of investigation.

## 4. RIG-I Like Receptors (RLRs)

### 4.1. RIG-I

Mammalian RIG-I is the founding member of RLRs and is a cytoplasmic sensor of viral dsRNA. Mammalian RIG-I consists of two N-terminal CARDs that are involved in downstream adaptor interaction and signal integration, and a central DExD/H-box helicase domain and a C-terminal domain (CTD) that promote dsRNA binding [108]. In resting state, CARD domains remain auto-repressed, whereas upon RIG-I binding of dsRNAs with 5′-triphosphate (5′ppp) moieties and/or ATP, the relieved CARDs undergo homotetramerization and homotypic interactions with CARD domains of downstream adaptor MAVS [42,108]. RIG-I-MAVS CARD-CARD interactions then trigger MAVS filament formation that function as signaling platforms to recruit downstream signaling molecules including TRAFs, TBK1, IKKε, and IRF3 [108]. These signaling platforms contribute to a final signaling event where IRF3/7 phosphorylation occurs, leading to their dimerization and nuclear translocation to upregulate type I IFNs and other antiviral effectors.

With regards to ligand specificity, RIG-I preferentially binds the 5′ppp end of dsRNAs < 1kb in length and undergoes translocation to the interior of RNA in an ATP-dependent manner [42,109]. RIG-I translocation exposes 5′-ppp end to allow subsequent iterations of 5′ppp end binding and translocation by remaining RIG-I molecules, ultimately resulting in filamentous oligomerization at the dsRNA end [108]. Filamentous oligomerization facilitates CARD interactions between adjacent RIG-I molecules and consequently promotes CARD tetramerization [108]. As RIG-I assembly and filament formation depends on end to interior translocation and with 5′-ppp ends of dsRNA becoming a limiting factor for long dsRNAs, RIG-I sensing is limited to dsRNAs of less than 1 kb in length [108]. Additional mechanisms that are required for RIG-I activation and signaling include RIG-I intra-filament interaction and K63-polyubiquitination by RIPLET (RNF135), resulting in inter-filament bridging (or higher order oligomerization) into RIG-I clusters. Finally, recent studies in mammalian cells uncovered the obligatory requirement for RIPLET but not TRIM25 E3 ligase in RIG-I clustering and activation [110]. RIG-I clustering and K63-polyubiquitin chains allow RIG-I CARD tetramer formation, which then promotes MAVS filament formation via CARD-CARD interactions.

Although RIG-I and its E3 ubiquitin ligase RIPLET are absent in chickens, duck (and goose) encodes a functional RIG-I orthologue that displayed 53% amino acid identity to human and 78% identity to zebra finch RIG-I [111,112]. Duck (and goose) RIG-I shares similar domain organization with mammalian RIG-I and consists of tandem N-terminal CARD domains, a DeXD/H box helicase domain, and a C-terminal regulatory domain. Duck (and goose) RIG-I is a dsRNA-dependent ATPase with conserved Walker A ATP-binding motif identical to mammalian RIG-I [112,113]. Although the hydrophobic core and the four lysine (K858/861/888/907) residues implicated in ligand-binding were found to be conserved in duck (and goose) RIG-I, residues (T55 and K172), critical for interaction and polyubiquitination by TRIM25 to facilitate MAVS binding and downstream signaling, were not conserved in duck (and goose) RIG-I [112]. In light of these observations, duck RIG-I activation and signaling was suggested to function in a different manner compared to mammals or involves different residues in avian [112]. On the other hand, recent studies identified RIPLET as an obligatory E3 ubiquitin ligase that ubiquitinates mammalian RIG-I at residues in tetramerization interface (K48, K115) and flexible C-terminus of 2CARD (K190, K193) to a minor and major extent, respectively [110,114]. Although multiple previous studies described RIG-I ubiquitination by TRIM25 in the context of isolated mammalian and duck RIG-I CARD domains or ectopically overexpressed full-length RIG-I, endogenous RIPLET but not TRIM25 was recently found to be required for RIG-I ubiquitination by multiple independent knockout cell lines stimulated with a variety of RIG-I ligands [110,113]. Although K193 is conserved in duck RIG-I and is a site of K63-linked ubiquitination, remaining residues are not conserved. Finally, ubiquitin binding sites (R71, D75, K95, and E98) important for RIG-I signaling in mammalians were conserved in avian RIG-I [115]. These observations point to RIPLET-mediated functions in RIG-I signaling in avian-like mammals in a ubiquitin-dependent or -independent manner. Unlike chicken, duck and goose encode functional homologs of RIPLET with RING domain that contributes to E3 ligase activity, coiled coil, and C-terminal SPRY domains.

Duck and wild waterfowl are the natural reservoirs of many influenza virus subtypes that remain asymptomatic, whereas avian influenza virus infection of chickens is lethal [112]. The higher resistance of ducks than chickens to HPAI-infections has been imputed to RIG-I-induced IFN-β and ISG induction [112]. In support of this, HPAI (H5N1)-infection of ducks induced dramatic upregulation of RIG-I in lungs, whereas this is not the case in LPAI (H5N2)-infected ducks. Furthermore, ectopic expression of duck RIG-I in DF-1 cells conferred RIG-I ligand responsiveness and profound antiviral effect towards HPAI infection with a significant reduction in viral titers and concomitant upregulation of antiviral genes including chicken *IFN-β, MX1, OASL PKR,* and *IFIT5* [116]. Interestingly, duck RIG-I displayed weaker IFN activating ability compared to goose RIG-I in transfected DF-1 cells with or without influenza virus infection [117].

New Castle’s disease virus (NDV) is an avian paramyxovirus that causes lethal disease in chickens, whereas ducks and sand geese are generally resistant to NDV. Transfection of DF-1 cells with goose RIG-I or RIG-I CARD conferred ligand responsiveness to 5′-ppp 21-mer RNA with an enhanced IFN-β promoter activity [117]. Furthermore, goose RIG-I transfected 293T/17 or DF-1 cells displayed an antiviral effect towards NDV infection and a significant upregulation of IFN-related genes *IRF-3 and IFIT1* [117]. Lastly, in a panel of mouse normal and tumor cell lines, NDV infection was inversely correlated with RIG-I expression, which may contribute to the oncolytic effect of NDV in RIG-I/IFN-deficient tumors [118]. MAVS protein functions as an adaptor downstream of RIG-I or MDA5 to form signaling platforms that ultimately lead to IRF3 activation and type I IFN production in response to dsRNA sensing during viral infections [119]. Chicken, duck, and goose encode a mammalian orthologue of MAVS that functions downstream of avian RIG-I (in case of duck and goose) and MDA5. Although avian MAVS displayed similar domain organization with human MAVS and contain a CARD domain, a proline-rich region, and a C-terminal TM domain, avian MAVS exhibit significantly less similarity to human MAVS (≈16%) than within avian species (55%) [120]. The CARD domain in general displayed an overall higher sequence conservation including residues T54, G67, W68, and V69, which were demonstrated to be critical of IFN-β induction by human MAVS, pointing to the essential function of CARD-CARD interactions in MAVS filament formation [120]. Interestingly, avian MAVS contained an expanded stretch of proline-rich region with numerous putative TRAF-binding motifs (107–173 aa in human, 103–297 aa in chicken, 103–302 aa in duck, 103–229 aa in goose). On the other hand, two 30aa deletion stretches were present in avian MAVS compared to human. None of the sites of phosphorylation (positive regulators, S442 and Y9; negative regulator, T234) and ubiquitination (438DLAIS442 motif, K362, K461, K468, K500) observed in human MAVS were conserved in avian MAVS, suggestive of a distinct regulation of avian MAVS [120]. Furthermore, phosphorylation of the pLxIS motif (p: hydrophilic residue; x: any residue) of human MAVS that is required for the recruitment and activation of IRF-3 is absent in avian MAVS [121]. In line with this, in contrast to human MAVs, goose and chicken MAVS failed to activate human IFN-β promoter in HEK293T cell line [122]. Interestingly, whereas ectopic overexpression of goose and chicken MAVS potently activated ISG expression in goose embryonic fibroblasts, goose MAVS failed to activate chicken IFN-β promoter in DF-1 cell line, indicative of species-specific signaling mechanisms operating downstream of MAVS [122]. Finally, ectopic overexpression of duck MAVS in duck embryo fibroblasts potently activated IFN-β promoter and siRNA knockdown of MAVS abrogated Sendai virus (SeV)-induced IFN-β promoter activation [122].

Cloning and characterization of duck MAVS revealed similar domain organization with mammalian MAVS and the closest evolutionary relationship to chicken [123]. Overexpression of MAVS in DEF cells activated NFκB and IRF1 promoters, and MAVS CARD and TM domains were critical for its activity [123]. Finally, poly I:C-induced IFN-β induction was MAVS-dependent, indicating conserved RIG-I-MAVS-TBK1-IRF7 signaling axes [123].

### 4.2. MDA5

Mammalian MDA5 shares similar domain organization and downstream signaling with that of RIG-I. MDA5 binds and forms filaments along the length of long dsRNA in a sequence-independent manner. In contrast to RIG-I, filament formation by MDA5 protomers is obligatory for high affinity interaction with dsRNA [124]. In addition, unlike RIG-I binding to dsRNA ends or 5′-ppp groups, MDA5 preferentially binds to long dsRNA interior and nucleates filament elongation from dsRNA interior in an ATP-independent manner [124]. It is presently unknown whether MDA5 undergoes additional levels of oligomerization such as filament cross-bridging, as seen with RIG-I. However, MDA5 K63-linked polyubiquitination relies on TRIM65 E3 ligase rather than RIPLET [125].

An amino acid comparison of chicken MDA5 to MDA5 from several other species displayed an ≈60% level of identity with the level of identity lower over the N-terminal 300 aa (≈37%) compared with the C-terminal region (≈70% identity) [126]. Chicken MDA5 (chMDA5) contains the typical DEXD/H and RIG-like helicase domains characteristic of mammalian MDA5 [126]. Initial studies investigated the ability of chMDA5 to compensate for the lack of RIG-I during influenza virus infection and concluded reduced IFN-β production in response to viral RNA compared to poly I:C-induced IFN-β. In addition, MDA5 was proposed to be insufficient to counter an influenza virus infection in chickens. Some peculiarities observed with chMDA5 include lack of length specificity for RNA ligands (unfractionated poly I:C, long (≈6 kb), medium (≈3 kb), or short (≈1 kb) RNAs all led to a similar level of IFN-β), unlike mammalian MDA5, and the characteristic ability to respond to 5′-ppp or dephosphorylated dsRNA in a similar manner, unlike RIG-I [126]. A follow-up study, however, reported that chMDA5 preferentially senses short poly(I:C) (0.2–1 kb) rather than the long poly I:C (1.5–8 kb) in DF-1 cell line [127]. Finally, although ectopic overexpression of chMDA5 elicited significant amount of IFNβ in DF-1 cells that effectively countered LPAI infection, it failed to counter HPAIV infection.

Muscovy duck MDA5 contains similar domain architecture as mammalian MDA5, with two CARD motifs in the N-terminal region (aa 13–94 and aa 110–197), a type III restriction enzyme domain (aa 300–488), a helicase conserved C-terminal domain (aa 719–802), and a C-terminal regulatory domain (aa 877–1000) bearing two Zn2+-binding regions (aa 883–886 and aa 938–941) and one RNA-binding loop (aa 921–930) [128]. Muscovy duck MDA5 is highly similar to the MDA5 of chickens (86.1% identity), and less similar to human (63.5%) and mice (63.2%) [128]. Ectopic overexpression of duck MDA5 CARD led to a significant induction of antiviral genes including *Mx, OASL, PKR, IL-2, IL-6, IFN-α*, and *IFN-γ* in an IRF7-dependent manner [128]. In addition, MDA5 CARD overexpression effectively reduced HPAI virus (H5N1; DK212) replication by several fold [128].

The goose MDA5 amino acid sequence displays high similarity with chicken MDA5 (87.5%) compared with human (63.4%) or mouse (62.7%) MDA5 but shares identical domain architecture [129]. Upon experimental infection of Qingyuan geese with HPAI virus, a significant induction of *MDA5* was observed in the spleen, brain, and lungs of infected geese, compared to control [129].

### 4.3. LGP-2

LGP-2 is a third RLR that consists of helicase and C-terminal regulatory domains but lacks tandem CARDs to initiate CARD-CARD interaction with MAVS; hence, it cannot initiate downstream signaling [130]. However, LGP2 modulated RIG-I and MDA5 signaling due to viral infections. In response to viral infections that activate RIG-I (Sendai virus, vesicular stomatitis virus, Japanese encephalitis virus, except influenza virus) and MDA5 (encephalomyocarditis virus, mengovirus), LGP2 positively regulated IFN-β production in an ATP hydrolysis-dependent manner [130]. Interestingly, LGP2 or its ATPase activity was found to be dispensable for the recognition of synthetic RIG-I (5′-ppp) and MDA5 (dsRNA) ligands and type I IFN production [130].

Similar to mammalian LGP-2, chicken, duck, and goose LGP-2 lack tandem CARDs and display similar functional effects [131,132,133,134]. Upon LPAIV A/duck/Hokkaido/Vac-1/04 (H5N1) infection of DF-1 cells overexpressing LGP2 or knocked down for endogenous LGP2, a reduced IFN-β promoter activity or IFN-β activity was observed [131]. In a similar vein, although duck MDA5-transfected DEF cells displayed an antiviral effect against duck enteritis virus (DEV) infection by enhanced induction of IFN-related genes (*Mx, OASL, IFITM1,* and *IFN-β*), LGP-2 had a concentration-dependent effect [132]. Higher concentrations of LGP-2 were found to reduce MDA5-induced IFN-β transcription [132].

## 5. Additional RNA Sensors

### 5.1. OASL

In humans, OASL belongs to the oligoadenylate synthetase (OAS) family proteins, which additionally include OAS1, OAS2, and OAS3. Although mouse comprise two OASL genes, *Oasl1* and *Oasl2*, which exhibit 74% and 49% identity with human OASL, respectively, Oasl2 forms the functional equivalent of human OASL [135]. In addition, mouse comprise eight Oas1 genes, and one gene each of Oas2 and Oas3. The OAS1-3 proteins exhibit significant homology to each other and only differ in the number of OAS domains. OAS1, OAS2, and OAS3 contain one, two, and three OAS domains, respectively [136]. One OAS domain of OAS1-3 proteins and mouse Oasl1 bear nucleotidyl-transferase (NTase) activity, which catalyzes the synthesis of 2′-5′ oligoadenylates (2′-5′A) upon viral dsRNA binding. Newly synthesized 2′-5′ oligoadenylates activate latent RNase L to result in cleavage of cellular and viral RNA and inhibition of protein synthesis. In addition, the catalytic activity of OAS1 and OAS2 relies on oligomerization into enzymatically active tetrameric and dimeric forms, respectively [136]. The conserved CFK motif in the OAS domain of OAS1 and OAS2 mediates OAS oligomerization and affects the synthesis of effective 2′-5′A [136]. OAS3 and OASL, on the other hand, lack CFK motif and therefore function as monomers [136]. The processivity of OAS enzymes is dependent on their ability to oligomerize, with OAS3 synthesizing 2′-5′A dimers, whereas OAS1 and OAS2 are capable of synthesizing 2′-5′A trimers and tetramers [136].

Human OASL contains one OAS domain that lacks NTase activity, a CCY motif instead of CFK in the OAS domain, and two ubiquitin-like (UBL) repeats in the C-terminus [136]. Thus, the OASL lacks the ability to oligomerize, 2′-5′A synthase activity and downstream RNAseL activation mechanism. Although mouse Oasl1 also lacks NTase activity, mouse Oasl2, unlike human OASL, contains two critical aspartic acidresidues in its active site and exhibits OAS enzyme activity [136]. OASL displays antiviral activity against RNA viruses through its C-terminal UBL domain [137]. Upon initial viral infection and OASL induction as an ISG in response to IFN signaling, OASL bound to RIG-I and mimicked K63-linked ubiquitin [137]. Typically, RIG-I activation requires dsRNA ligand and K63-linked ubiquitination by RIPLET to undergo intra filament and inter filament bridging. OASL-mediated activation of RIG-I is more sensitive in that RIG-I requires only dsRNA ligand for its activation [137]. Recently, OASL was found to negatively regulate cGAS-STING signaling and type I IFN production upon dsDNA virus infection [138]. Mechanistically, OASL was found to directly interact with cGAS to inhibit its enzymatic activity, and thus cGAMP production. OASL-cGAS interaction was found to be independent of dsDNA, and unlike RIG-I-OASL interaction, did not require C-terminal UBL domain [138]. OASL-mediated negative regulation of cGAS activity and type I IFN production was proposed to be a feedback mechanism to limit exuberant type I IFN production during chronic DNA viral infections [138]. The mouse Oasl1, on the other hand, also functions as a negative regulator of IFN induction, although in a different manner. Oasl1 inhibits the translation of IRF7 through an interaction with the stem loop structure in the 5′-untranslated region of IRF7 mRNA. Negative regulation of type I IFN production by mouse Oasl1 promotes viral persistence and represses T-cell function.

Compared to mammals, avians possess a contracted OAS gene family with a single member (OASL) that functions in an RNase L-dependent and -independent manner. Avian OASL possesses an N-terminal NTase domain; one OAS domain and two tandem C-terminal UBL domains; and several conserved loops or motifs such as LxxxP motif important for the synthetase activity, the P loop involved in the ATP binding activity, the D-D Box involved in Mg2+ binding, a lysine/arginine-rich region (KR-RR) that facilitates dsRNA binding, and a CFK motif to promote oligomerization of OASL [139]. Chicken, duck, and goose OASL share similar domain organization, as described above. However, functional characterization indicated that avian OASL has accrued species-specific adaptations and functional divergence compared to their mammalian counterparts [137]. For instance, unlike mammalian OASL (human OASL and Oasl2), duck OASL binds dsRNA with an enhanced affinity and activates RNase L system in a UBL-dependent manner [137]. In the absence of UBL domains, duck OASL binds dsRNA weakly and synthesizes 2′-5′A but fails to activate RNAse L [137].

Chicken OASL (ChOASL) gene encodes two alternatively spliced forms, ChOASL-A and ChOASL-B, that are 508 and 476 amino acids in length, respectively [140]. ChOAS-B has a 32 aa deletion in the UbL1 portion and exhibits reduced dsRNA binding and 2′-5′A synthetic activity compared to ChOASL-A [141]. These observations further support the idea of species-specific adaptation of avian OASL in which UBL domain contributes to dsRNA binding, and thus effective synthesis of 2′-5′A. Furthermore, in contrast to duck OASL (duOASL), deletion of two C-terminal UBL domains of chOASL-A led to abrogation of 2′-5′A synthesis, indicating that the impact of UBL domains on dsRNA binding affinity is much higher with chOASL compared to its duck counterpart [142]. In addition, when conserved D residues were mutated to abolish 2′-5′A synthesizing activity and RNase L activation, duOASL antiviral activity switched to activate RIG-I [137]. On the other hand, mutation of conserved lysine/arginine residues that promote dsRNA binding activated neither the OASL/RNase L pathway nor the OASL/RIG-I pathway [137]. Finally, duck OASL exhibited broader antiviral activity against two strains of double-stranded RNA viruses (infectious bursal disease virus, IBDV/B87, and respiratory enteric orphan virus, REOV/Z97/C10), and several negative stranded RNA viruses including one H1N1 (A/Puerto Rico/8/34, PR8), three H5N1 viruses (A/duck/Hubei/49/05, DK/49; A/goose/Hubei/65/05, GS/65; A chicken/Hubei/0513/2007, CK/0513), and a New Castle’s disease virus (La Sota strain) [137]. No antiviral activity against a DNA virus (Fowlpox virus) was observed [137].

Interestingly, functional characterization experiments of goose OASL (goOASL) performed in duck embryo fibroblasts (DEFs) demonstrated that antiviral activity of goOASL against duck tembusu virus (DTMUV) is independent of both 2′-5′A synthetase activity and UBL-domains [139,143]. These findings led the authors to conclude that goOASL possess additional RNase L and RIG-I-independent antiviral mechanisms [139,143]. However, the DEFs the authors employed in the study may still contain endogenous duOASL, which may complement the loss of function of goOASL mutants used in that study, and therefore these findings beg confirmation.

### 5.2. PKR

Protein kinase R is a dsRNA sensor that is expressed at lower levels in normal cells but highly expressed in IFN-stimulated cells as an ISG. Upon viral dsRNA binding, activated PKR is a multifunctional protein that regulates translation, apoptosis, stress responses, metabolism, and NFκB/IRF3-dependent and -independent signaling [144]. The domain architecture of PKR consists of an N-terminal dsRNA binding domain composed of two tandem repeats of a conserved dsRNA binding motif 1 and 2, (dsRBM 1 and 2) interspaced by a linker and followed by a flexible linker connected to a C-terminal kinase domain [144]. Studies with mammalian PKR demonstrated that both the dsRBMs are essential for high affinity dsRNA binding [145]. Upon dsRNA binding, activated PKR primarily induces phosphorylation of eukaryotic initiation factor 2α (eIF2α) at serine 51, resulting in a halt of translation of cellular and viral mRNAs [146]. This is facilitated by interaction of PKR with eIF2α by a specific alpha-helix unique to PKR (aG), which is located on the surface of the C-terminal lobe of the kinase domain [146].

Although chicken and goose PKR genes and functions have been characterized, duck PKR functions are yet to be studied [147,148]. Both chicken and goose PKR display similar domain structure as mammalian PKR with 3 main domains, 2 dsRBMs, and 1 serine/threonine protein kinase domain comprising 11 kinase sub domains [147,148]. Interestingly, duck PKR comprises only two domains, dsRBM 1 and a serine/threonine protein kinase domain [148]. Overexpression of chicken PKR in 3T3 fibroblasts and goose PKR in GEFs displayed antiviral effect against vesicular stomatitis virus carrying GFP and NDV, respectively, with a significant reduction in viral titers [147,148].

### 5.3. DDX1, DDX21, and DHX36 Complex

In mammalian cells, in response to poly I:C stimulation, TRIF deletion led to reduced IFN production compared to TLR3 deletion, suggestive of TLR3-independent but TRIF-dependent IFN production mechanisms [149]. D2SC cell lysate proteins pulled down by biotinylated poly I:C precipitated three members of the DExD/H helicase protein family, DDX1, DDX21, and DHX36, in addition to well-known interactors such as RIG-I, MDA5, and LGP2 [149]. Upon further investigation, DDX1, DDX21, and DHX36 complex was identified as a bona fide dsRNA sensor that bound both short and long poly I:C [149]. Specifically, DDX1 helicase A domain bound poly I:C, whereas the SPRY domain of DDX1 bound the PRK domain of DDX21. The PRK domain of DDX21 bound the Helicase C-HA2-DUF domains of DHX36 to form a bridging interaction between DDX1 and DDX36 [149]. The TIR domain of TRIF interacts with DDX21 and DHX36. Downstream, DDX21 and DHX36 interact with adaptor TRIF via their PRK and HA2-DUF domains, respectively, to promote downstream activation of NFκB and IRF3, resulting in type I IFN production [149]. Finally, poly I:C stimulation enhanced the interaction between downstream TRIF and MAVS. DDX1, DDX21, and DHX36 knockdown caused a 60–70% reduction in IFN-β production and a 40–50% reduction in TNF-α production by D2SC cells upon influenza A virus, whereas a 70–80% reduction in IFN-β production and a 60–70% reduction in TNF-α production was observed in D2SC cells upon reovirus infection [149].

Chicken, duck, and goose encode functional orthologs of mammalian DDX1, DDX21, and DHX36 that displayed ≈98-100% aa identity with human counterparts. Furthermore, functional motifs associated with DDX1 (SPRY, DEAD, HelicC), DDX21 (PRK, DEAD, HelicC, GUCT), and DHX36 (COG, HelicC, HA2 and DUF) complex interactions with each other and with TRIF were conserved in avian DDX1/DDX21/DHX36, suggestive of functional interactions (Appendix A). Species-specific features included a 43 aa deletion stretch comprising lysine rich region in N-terminus of avian DDX21 and deletions in the N-terminus of avian DHX36. Functional characterization of DDX1/DDX21/DHX36 complex interactions with dsRNA ligand and downstream signaling in avian awaits further investigation.

An independent function of DHX36 in dsRNA sensing and enhancing the activation of another dsRNA sensor, protein kinase R (PKR), was demonstrated in mouse embryonic fibroblasts [150]. PKR activation in response to viral dsRNA sensing results in its autophosphorylation and phosphorylation of eukaryotic initiation factor 2 alpha (eIF2α) on serine 51 to inhibit the initiation step of cellular and viral mRNA translation [150]. Failure to initiate translation promotes nucleation of initiation stalled mRNAs into messenger ribonucleoprotein complexes known as stress granules, which serve as innate immune platforms and have antiviral functions [150]. In this regard, DHX36 moderately enhanced the binding of PKR to dsRNAs, as well as promoted the activation of PKR through its ATPase activity ultimately resulting in the formation of stress granules. Antiviral functions of stress granules and relevance of DHX36/PKR interactions in avian species remains to be investigated.

### 5.4. DDX3

DDX3 helicase is critical for development as mice lacking DDX3 display prenatal lethality [151]. In addition, DDX3 is essential for hemopoiesis and maintenance of B cells and NK cells, whereas macrophages, neutrophils, or CD11c+ DCs showed little dependence on DDX3 [152]. As a sensor, DDX3 is known to bind dsRNA to supplement RIG-I and MDA5 signaling. In this regard, DDX3 was reported to associate with MAVS to potentiate downstream induction of type I IFN [153]. Furthermore, DDX3 interacts with TBK1 as a substrate and also with IKKε to facilitate phosphorylation of IRF3/7 and type I IFN gene transcription [154]. In support for this, DDX3 facilitates enhancement of type I IFN induction by constitutively active IRF7 in cells lacking TBK1 [152]. Henceforth, cells lacking DDX3 are severely deficient in the production of type I/II IFN and pro-inflammatory cytokines including IL-1β, IL-6, Nos2, and CCL5 [152]. Finally, in a mouse challenge model of *L. monocytogenes*, DDX3 deficiency displayed selective defects in hematopoiesis, which hampered the ability to establish the inflammatory milieu needed for innate defense. This includes impaired antimicrobial gene expression in macrophages and a lack of IL-12-IFNγ axis necessary for protection [152].

Chicken, duck, and goose encode a functional ortholog of DDX3 that exhibits high identity to mammalian DDX3 with well-conserved catalytic helicase core domain, whereas the N-terminal or the arginine-serine-rich (RS) region in the C-terminal domain were poorly conserved in goose DDX3 (Appendix A) [155]. Interestingly, chicken DDX3 was found to interact with STING to mediate type I IFN production via STING-TBK1-IRF7-IFN-β signaling axis. Finally, DDX3 overexpression induced an antiviral effect against LPAI and NDV infection in DF-1 cells [155].

### 5.5. DDX23

Human DDX23 is an evolutionarily conserved nuclear dsRNA sensor that binds low molecular weight poly I:C via its N-terminal region and undergoes cytoplasmic translocation. In the cytoplasm, DDX23 interacts with TRIF or MAVS adaptor via its DExD/H domain to form DDX23-TRIF and/or DDX23-MAVS interactions and initiates downstream signaling resulting in NFκB and IRF7 activation and type I IFN production [156]. DDX23 knockdown in A549 cell line led to reduced production of IFN-β, IL-6, and RANTES in response to poly I:C stimulation and enhanced the replication levels of VSV-GFP virus.

Chicken, duck, and goose encode true ortholog of mammalian DDX23 that display ≈96% aa identity with human DDX23. In terms of sequence conservation, both DExD/H and helicase C-terminal domains display a greater degree of identity compared to the N-terminal region (Appendix A). In addition, avian DDX23 displayed several conserved putative bipartite NLSs in the N-terminal region, pointing to similar nuclear localization to that of mammalian DDX23. Functional characterization of avian DDX23 as a functional dsRNA sensor awaits investigation.

### 5.6. DDX24

In mammalian cells, DDX24 was found to be an ISG that negatively regulated RLR signaling, including both RIG-I and MDA5 induction of type I IFN production in an RNA-dependent and -independent manner [157]. In addition to binding and sequestering RLR ligands, DDX24 was found to inhibit RIP1-mediated activation of IRF7, downstream of RLR signaling. In unstimulated MEFs, DDX24 displayed nucleolar localization, whereas in poly I:C-stimulated MEFs, DDX24 relocated to cytoplasm and interacted with FADD and RIP1 to prevent IRF7 activation [157].

Chicken, duck, and goose encode true ortholog of mammalian DDX24 that display ≈91% aa identity with human DDX24, with a greater degree of conservation in the C-terminus than N-terminus (Appendix A). DDX24 localization in unstimulated avian cells and relocation upon poly I:C stimulation and IFN regulation must be a subject of future investigation.

### 5.7. DDX60

Although DDX60 is not a bona fide RNA sensor, DDX60 was demonstrated to function as a sentinel for RIG-I activation in addition to RIG-I-independent antiviral function [158]. In mouse embryonic fibroblasts, peritoneal macrophages, and splenic CD11c + cells, RIG-I-mediated type I IFN production requires DDX60, whereas this is not the case in BM-DCs, suggestive of cell-type-specific roles of DDX60. In support of this, RNA virus-mediated immune evasion of DDX60 function was demonstrated, where EGFR signaling activation by influenza A and hepatitis C viruses led to phosphorylation of DDX60 and impaired RIG-I responses [158]. RIG-I-independent antiviral function of DDX60 includes viral RNA degradation in an exosome-dependent manner [158].

Avian, including chicken, duck, and goose, encode a functional ortholog of DDX60 that displays 97% aa identity with its human counterpart (Appendix A). In addition to a conserved lysine 791 involved in ATP binding, critical sites of phosphorylation Y793 and Y796 were also found to be conserved in avian lineages, indicative of mechanisms of regulation identical to that of mammalians.

### 5.8. Zinc Finger NFX1-Type Containing 1 (ZNFX1)

Recently zinc finger NFX1-type containing 1 (ZNFX1) was identified as a long dsRNA sensor that functions to sense viral RNA during the early stages of viral infection [159]. In A549 human lung adenocarcinoma and mouse embryonic fibroblast cells, ZNFX1 was found to display lower expression under steady state conditions but higher expression than the canonical dsRNA sensors RIG-I and MDA5. Under ligand- (HMW poly I:C) and IFN-stimulated conditions, ZNFX1 expression is drastically enhanced as an ISG [159]. ZNFX1 belongs to the SF1 helicase family that is highly conserved from fruit fly to human and contains an armadillo-type fold (ARM) domain, a P-loop helicase domain, and a zinc finger (ZF) domain [159]. ZNFX1 is a mitochondrion localized as a peripheral membrane protein that, upon sensing long dsRNA (or HMW poly I:C) through its ARM and P-loop domains, interacts with MAVS via ARM domain and promotes downstream induction of type I IFNs and ISGs in a RIG-I/MDA5-independent manner [160]. ZNFX1 is thus a non-redundant dsRNA sensor.

Chicken, duck, and goose ZNFX1 display similar domain architecture as human ZNFX1 in terms of ARM, P-loop, and ZF domains (Appendix A). Functional characterization of avian ZNFX1 is a subject of future investigation, and whether avian ZNFX1 prefers long dsRNA to induce type I IFN responses remains to be studied.

## 6. DNA and Cyclic Dinucleotide (CDN) Sensors

### 6.1. cGAS-STING DNA Sensing Pathway

CDNs have recently emerged as critical ligands in the innate immune response to bacterial and viral infections in mammals. Like mammals, the importance of c-GMP-AMP synthase (cGAS)-mediated DNA sensing and activation leading to 2′-3′-cGAMP production in response to viral infections is well described in avian. 2′-3′-cGAMP serves as a ligand to the ER TM adaptor known as STING that, upon ligand binding, undergoes dimerization and oligomerization, leading to its activation and downstream induction of type I IFNs, contributing to an antiviral state [161,162]. In addition to host-derived CDNs, bacteria also produce CDNs such as c-di-GMP, c-di-AMP, and 3′-3′-cGAMP that serve as second messengers to regulate a variety of bacterial processes [162,163,164,165]. For instance, c-di-GMP regulates bacterial motility, virulence, and biofilm formation in a wide range of bacterial species, whereas 3′-3′-cGAMP was found to regulate metal-reducing activity and host colonization in *Geobacter* and *Vibrio cholera*, respectively. c-di-AMP is an essential CDN detected in many bacterial species and regulates bacterial growth, cell wall homeostasis, and metabolism [160,161,162,163,165]. Recent studies in mice and humans identified bacterial CDN sensors, RECON, and ERADp that, upon binding bacterial c-di-AMP, lead to downstream induction of pro-inflammatory cytokines and establishing an antibacterial state [161,162].

#### 6.1.1. cGAS

The nucleotidyl transferase enzyme cGAS is a cytosolic foreign DNA sensor that is activated upon direct binding to DNA in a sequence-independent manner [166]. DNA binding with ladder-like networks of cGAS dimers trigger conformational changes to induce cGAS enzymatic activity [167]. Activated cGAS synthesizes endogenous second messenger 2′-3′ cGAMP through GTP and AT P [168]. 2′-3′ cGAMP binding to the endoplasmic reticulum-localized adaptor known as STING triggers conformational changes resulting in STING closed confirmation and oligomerization. In resting state, ER retention of STING is mediated through interaction with the Ca2+ sensor stromal interaction molecule 1 (STIM1) [169]. Following cGAMP binding-induced conformational changes, STING traffics through ERGIC and the Golgi apparatus, where it interacts with TANK-binding kinase 1 (TBK1), resulting in its direct phosphorylation by TBK1. Finally, the C-terminal tail region of STING serves as a docking site for IRF3, which is then phosphorylated by TBK1, resulting in its activation, dimerization, and nuclear translocation to regulate the transcription of type I IFNs [170].

Human cGAS consists of an N-terminal tail (amino acids 1–160), which permits nuclear accumulation, centromeric association, and activation of cGAS in cycling cells [171]. Comparatively, chicken, duck, and goose cGAS were found to exhibit shortened N-termini with least similarity of 22.4%, 17.4%, and 7.7% to human cGAS, respectively [172]. Chicken and duck cGAS N-termini, unlike goose cGAS, however, appeared to be rich in positively charged amino acids, suggestive of functions associated with nuclear localization and DNA binding. The nucleotidyl transferase (NTase) domain (residues 161–512 in human cGAS) was found to be necessary and sufficient for DNA recognition and cGAMP production [166]. The NTase domain consists of α/βcore, helix bundle, spine helix, activation loop, and zinc finger domains. The NTase domains of chicken, duck, and goose exhibited a greater level of similarity to human cGAS, with amino acid similarities of 69.2%, 69.3%, and 55.8%, respectively. One peculiarity observed with goose cGAS despite a severely shortened N-terminus is the lack of spine helix and instead an extended C-terminal tail of 21 amino acids [166].

Human cGAS NTase domain adopts the typical NTase fold with a bilobed architecture comprising an N-terminal α/β catalytic core and a C-terminal helix bundle [166]. Similar to human cGAS, conserved catalytic residues (glutamate (Glu) 225, aspartate (Asp) 227, and Asp319 in humans) are located on the centrally twisted β-sheets of the α/β core, and catalytic pocket is formed between the α/β core and the helix bundle in avian (Appendix A) [166]. Furthermore, the activation loop in the vicinity of the catalytic pocket (residues 210–220 in humans) that contributes to nucleotide recognition and crucial for the catalytic activity of cGAS also appeared to be conserved in chicken, duck, and goose cGAS. The long N-terminal helix, also referred to as the spine helix, and the CCHC-type zinc finger exist on the opposite side of the catalytic pocket and are critical for DNA recognition in human cGAS. The spine helix contributes to dsDNA-specific recognition and the conformational activation of cGAS, whereas the zinc finger contributes to cooperative DNA binding and the formation of the 2:2 dimer complex, resulting in the activation of cGAS and cGAMP production. Upon dsDNA binding, the leucine (Leu) residue on the spine helix (Leu174 in human cGAS) relocates from the solvent toward the catalytic pocket, thus stabilizing the activation loop and rearranging the catalytic residues to form the catalytic pocket [166]. Although goose cGAS lacked the spine helix, spine helix of chicken and duck cGAS contained conserved leucine, indicating similar mechanisms of dsDNA binding and catalytic site rearrangement to human cGAS (Appendix A). The lack of spine helix in goose cGAS points to its N-terminus in carrying out dsDNA recognition and conformational activation functions. In this regard, fifth leucine residue in goose cGAS may contribute to activation loop stabilization and catalytic pocket rearrangement.

Like in mammalian, cGAS-STING-TBK1-IRF7 axis plays a critical role in restricting DNA virus infection in avian. However, an avian tumor virus known as Marek’s disease virus has evolved interferon-antagonistic proteins such as VP23 and oncoprotein Meq which suppress DNA sensing pathways and downstream IRF7 activation [173,174].

#### 6.1.2. STING

ER adaptor STING binds 2′-3′ cGAMP to activate downstream innate immune signaling. The full-length human STING consists of a tetraspanning TM region (TM1-4) that positions it in the ER membrane, a cyclic dinucleotide binding domain (CDN), and a C-terminal tail [166]. Studies on human STING mapped the interaction interface between CDN and the ligand-binding pocket of STING, which was divided into the base, ribose, and phosphate recognition sites. The CDN bases were found to be recognized by side chains of Y167 and R238 (human STING numbering), whereas the edges of the bases are recognized by water-mediated hydrogen bonding with the lid amino acids V239 and S241, allowing purine-specific recognition of CDN. Finally, the phosphate moieties and hydroxyl groups of ribose moiety were recognized by R238 salt bridge and hydrogen bonding side chains of T267 and S162, respectively [166].

Sequence alignment of chicken, duck, and goose STING showed amino acid identities of 43.4%, 42.34%, and 32.5%, respectively, to human STING (Appendix A). Despite low identity, amino acids critical for the recognition of various CDN moieties remained conserved, indicative of similar mechanisms of CDN recognition and STING activation. Finally, previous studies revealed pLxIS motif in the C-terminal tail of human STING, which is recognized and phosphorylated by TBK1 to elicit downstream signaling by binding IRF3 [121]. Notably, C-terminal tails of chicken, duck, and goose STING bore pLxIS motif, pointing to conserved actions of TBK1-mediated phosphorylation and binding of IRF7 (IRF3 is absent in avians) to promote downstream signaling (Appendix A) [175]. In line with this, deletion of pLxIS motif (SLQxSyS motif) in the C-terminal tail of chicken STING led to failure to activate downstream IRF7 and IFN-β promoter [175].

Functional characterization of duck STING in BHK21 or DEF cells revealed its localization in the ER and mitochondria. Furthermore, STING overexpression led to significant activation of NFκB, IFN-β, and ISRE promoter elements along with an antiviral effect towards DPV infection [176]. Another study reported the antiviral effect of overexpressed STING against H9N2 LPAI virus infection in DEFs [177]. Lastly, DTMUV NS2B3 protease was demonstrated to cleave STING and abrogate downstream signaling to TBK1 signifying virus-encoded evasion mechanisms to subvert IFN induction [178]. Functional characterization of goose STING is pending and may have similar functions.

#### 6.1.3. ERADp

Studies in mice have decoded the sensor function of the ER membrane adaptor, ERADp, which senses and binds bacterial second messenger c-di-AMP [161]. Upon c-di-AMP binding, ERADp underwent dimerization and interaction with TAK1, resulting in TAK1 activation and downstream induction of pro-inflammatory cytokines IL-6 and TNF [161]. Interestingly, ERADp-dependent cytokine induction was found to be independent of cGAS-STING signaling [161]. In contrast to STING-dependent interferon induction, ERADp activation primarily results in pro-inflammatory cytokine production, as seen during bacterial infections. In support of this, ERADp exhibited higher affinity towards c-di-AMP compared to 2′-3′-cGAMP (Kd:76nM versus 2235nM). Conversely, STING ligand 2′3′-cGAMP exhibited higher affinity towards STING compared to c-di-AMP (Kd: 51nM versus 1853nM) [161]. In view of its critical function in sensing bacterial infections to induce immunity, ERADp protein homologs of chicken, duck, and goose exhibited the greatest similarity with human ERADp of 99.2%, 98.4%, and 92.8%, respectively. Furthermore, in light of the essential functions of the two TM domains and the C-terminal tail regions in the c-di-AMP sensing, TAK1 activation and pro-inflammatory cytokine induction, we noted greatest amino acid identities in these regions, pointing to conserved functions (Appendix A).

### 6.2. DDX41

DDX41 is a DEAD box protein family member that was found to recognize B-form DNA, and bacterial second messengers (c-di-GMP, c-di-AMP) [179,180]. In response to infection of mouse BMDCs and D2SC (mouse DC cell line) cells with *L. monocytogenes* and HSV-1, respectively, DDX41 mediated type I IFN, ISG, and pro-inflammatory cytokine (IL-6, TNFα) production in a STING-dependent manner [179,180]. DDX41 sensing of B-form DNA, c-di-GMP, and c-di-AMP is via its DEAD box [179,180]. Notably, deletion of one or both Walker motifs in DEADc domain prevented DDX41 interaction with DNA or STING, whereas ectopic expression of DDX41 lacking the HELICc domain led to greater activation of the *Ifnb* promoter than did the expression of full-length DDX41 [179]. Co-expression of DDX41 and STING led to synergistic induction of IFN-β compared with either protein expressed alone [40].

Cloning and characterization of chicken DDX41 revealed poly dA:dT ligand responsiveness, interaction with STING, and downstream activation of IFN-β promoter by overexpressed DDX41 such as mammalian DDX41 [181]. Duck DDX41 was also cloned and characterized in DEF cells, where it displayed cytoplasmic localization and interaction with STING, as observed in mammalian cells [182]. Furthermore, ectopic overexpression of DDX41 activated NFκB, IFN-β, and IRF1 promoter elements, along with an antiviral effect against dsDNA virus (DEV) infection. Apart from similar ligand (poly dA:dT) responsiveness to mammalian DDX41, duck DDX41 lacking the HELICc domain induced greater activation of the NFκB, IFN-β, and IRF1 promoters [182]. An overall greater sequence conservation was noted in avian DDX41 similar to human pointing to conserved functions (Appendix A).

### 6.3. DHX36

Apart from its role as a dsRNA sensor as part of the DDX1-DDX21-DHX36 complex, DHX36 was identified as a PRR of CpG-A (ODN2216) DNA in the primary human plasmacytoid dendritic cells (pDCs) and pDC cell lines (Gen2.2., Namalwa) [183]. DHX36 binds CpG-A via its DEAH domain and signals downstream in a MyD88-dependent manner. DHX36 interacts with the TIR domain of MyD88 through HA2 and DUF domains to result in IRF7 nuclear translocation and type I IFN production [183]. Furthermore, knockdown of DHX36 in Gen2.2 cell line resulted in a significant downregulation of IFN-α production in response to HSV-1 infection, whereas no effect was observed in regard to pro-inflammatory cytokine production or RNA virus infection [184]. These findings point to a role of DHX36 in sensing DNA viruses that is independent of the DDX1-DDX21-DHX36 complex, which senses dsRNA as described above. Functional characterization of avian DHX36 CpG-B sensing and downstream signaling is pending investigation.

### 6.4. DHX9

In human pDCs and pDC cell lines, DHX9 was identified to interact with CpG-B (ODN2006) via its DUF domain, which culminated in nuclear translocation of NFκB p50 subunit and downstream induction of pro-inflammatory cytokines (IL-6, TNFα) in a MyD88-dependent manner. The domains of DHX9 identified to be critical for interaction with MyD88 TIR domain included HA2, DUF, and HelicC domains. In response to HSV-1 infection of Gen2.2 cell line, DHX9 mediated pro-inflammatory cytokine production, whereas no effect of DHX9 knockdown on type I IFN production was observed. No true homolog of DHX9 is present in avian species [184].

### 6.5. hnRNPA2B1

In mouse macrophages, hnRNPA2B1 was recently demonstrated to sense naked but not nucleosome-wrapped DNA to initiate type I IFN production [185]. Mechanistically, upon sensing and binding HSV-1 DNA ejected into the nucleus, hnRNPA2B1 forms a homodimer, which is then demethylated by JMJD6 demethylase at R226. Demethylated hnRNPA2B1 is consequently translocated to the cytoplasm, where it activates the TBK1–IRF3 pathway in a Src kinase- and STING-dependent manner. In contrast, hnRNPA2B1 is not involved in NFκB-dependent induction of pro-inflammatory cytokines IL-6 and TNF. Interestingly, hnRNPA2B1 mutants (P81, K82, R83, V172, R173, K174) abrogated for dimerization due to mutations in the dimerization interface of RNA recognition motif (RRM) domain failed to translocate to cytoplasm and induce type I IFNs, whereas demethylated mutant (R226A in RGG domain) resulted in higher amounts of type I IFN production. On the other hand, methylated hnRNPA2B1 was found to function as a m6A modulator to promote the m6A modification and nucleocytoplasmic trafficking of *cGAS, IFI16,* and *STING* mRNAs in response to HSV-1 virus infection, resulting in their enhanced protein levels late during infection. Thus, pronged functions of hnRNPA2B1 facilitate initiation and enhancement of type I IFN production during early and late phases of nuclear DNA virus infection, respectively [185].

Avian and mammalian hnRNPA2B1 display remarkable conservation at the amino acid level, and whether avian hnRNPA2B1 also functions as a nuclear dsDNA sensor remains to be investigated (Appendix A).

## 7. Conclusions

Defining species-specific differences in PRR sensing and signaling is of critical importance, as PRR agonists contain immune modulatory activity and thus can be employed as vaccine adjuvants. Unique properties of avian PRRs in sensing and responding to PRR agonists must be considered to permit rational design of adjuvants for use in avian. For instance, LPS derivatives such as monophosphoryl lipid A (MPL) are widely used as vaccine adjuvants in mammals against papilloma virus and hepatitis B [186,187]. LPS derivatives act on TLR4/MD2 complex to promote type I IFN production through a MyD88-independent pathway [188,189,190]. *In ovo* administration of either free or PLGA-encapsulated LPS in embryonic day 18 (ED18) white leghorns resulted in a significant induction of IFN-γ, IL-1β, and IL-10 in spleen; IL-1β in lungs; and IFN-γ and IL-1β in the bursa of Fabricus [191].

One potential vaccine adjuvant for use in avian is CpG DNA which is recognized by TLR21. In chickens, CpG-ODN administration, along with traditional vaccine regimens, was found to boost the vaccine efficacy via adjuvant effects. This adjuvant effect of CpG when delivered via various routes, including *in ovo* route in embryos, was demonstrated against a wide range of bacterial, viral, and protozoal infections [72,191,192,193,194,195]. In commercial broilers, *in ovo* CpG administration via carbon nanotubes or liposomes was found to protect day-old chicks (60% protection) from *E. coli* or *Salmonella typhimurium* challenge [196]. *In ovo* administration of either free or PLGA-encapsulated synthetic class B CpG ODN 2007 in ED18 white leghorns resulted in a significant induction of IFN-α, IFN-γ, and IL-1β in spleen and lungs [191]. Furthermore, twice administration of encapsulated CpG once *in ovo* and again on day 14 post-hatch upon MDV challenge (on day 5 post-hatch) resulted in the lowest tumor incidence compared to twice-administered encapsulated LPS or LPS + CpG [193]. Also, *in ovo* administration of encapsulated synthetic class B CpG ODN 2007 in combination with HVT resulted in significant reduction in very virulent MDV-induced tumor incidence compared to HVT alone [194].

*In ovo* administration of class B CpG ODN 2007 in ED18 SPF embryos resulted in increased numbers of KUL01+ macrophages, IgM+ B cells, and CD4+ and CD8α+ cells in lungs of chicks on the day of hatch compared to non-CpG DNA or saline-inoculated embryos [197]. Furthermore, an increase in cells of adaptive immune system was noticed in spleen and bursa of hatched chicks. The increased cell recruitment correlated with a reduction in morbidity, mortality, and ILTV cloacal excretion due to ILTV challenge [197]. However, no reduction in oropharyngeal ILTV loads was noticed. In a similar vein, *in ovo* administration of class B CpG ODN 2007 also reduced mortality, morbidity, and oropharyngeal and cloacal viral loads upon infectious bronchitis virus (IBV; mass strain) challenge [198]. In parallel with reduced IBV N antigen expression in trachea, increased recruitment of KUL01+ macrophages in trachea, and CD4+ or CD8α+ cells in trachea and lungs was described [198].

CpG with a stable phosphorothioate backbone when administered via oral or intravenous or subcutaneous routes was found to reduce *Eimeria* oocyst shedding and promote weight gain in susceptible 3-week-old TK line of chickens. This was due to a proposed mechanism of T helper 1-mediated IL-12 and IFN-γ production [199,200]. In addition, CpG-ODN was proven to enhance the protection mediated by LaSota NDV vaccine and LPAI H4N6 virosomes by enhancing both Ab- and cell-mediated immunity [196]. Finally, other studies demonstrated the ability of CpG-ODN-mediated adjuvant effects during IBV and ALV-J vaccination [201,202,203].

Another mammalian PRR that recognizes DNA in a sequence-independent manner is cGAS, resulting in type I IFN production in a STING-dependent manner. Although cGAS stimulation by DNA results in type I IFN production in chicken, CpG-ODN stimulation of TLR21 was found to up-regulate the expression of INF-γ, IL-10, and IL-12p40, but not IFN-α and IFN-β [50,174]. DNA stimulation of cGAS can be harnessed to enhance anti-viral type I IFN production upon vaccination against DNA or RNA viruses.

Likewise, TLR3 agonist (poly I:C) enhanced the protective effects of sub-optimal HVT vaccine against very virulent MDV challenge [204]. Co-stimulation of chicken monocytes with TLR3 and TLR21 agonists synergistically upregulated Th1-cytokine IFNγ and regulatory cytokine IL-10 with an overall Th1-biased immune response [50]. The contribution of different PRRs including RIG-I, MDA-5, TLR3, and ZNFX1 in poly I:C-mediated induction of IFNs need to be investigated in avian. For this, CRISPR-mediated technologies can be employed to investigate the individual contribution of each PRR.

In a similar manner, the TLR5 agonist, bacterial flagellin, can also be employed as a potential adjuvant. Flagellin administration successfully reduced *S. enteritidis*-induced mortality in chickens by heterophil-mediated protective effects [205]. Adjuvant effects of monomeric and polymeric forms of *Salmonella* flagellin in combination with 64CpG adjuvant was evident in specific-pathogen free (SPF) chickens immunized either intramuscularly or intranasally with formalin-inactivated avian influenza (H5N2) virus vaccine [206].

Whole genome sequencing and comparative genome studies have enabled rapid identification and functional dissection of the avian PRRs to provide a better understanding of the avian innate immune system and mechanisms of disease resistance. In addition, PRR biology provides invaluable insights in the evolution of the vertebrate immune system. Despite significant progress being made, several properties of avian PRRs, both at the level of ligand specificity and downstream signaling, needed to be delineated. This knowledge allows rational design and use of immuno-modulatory agents in avian, and for a search for desired natural variations in PRR function among breeds. A major challenge is to translate the concepts of PPR biology into the complex in vivo setting.

## Figures and Tables

**Figure 1 vetsci-07-00014-f001:**
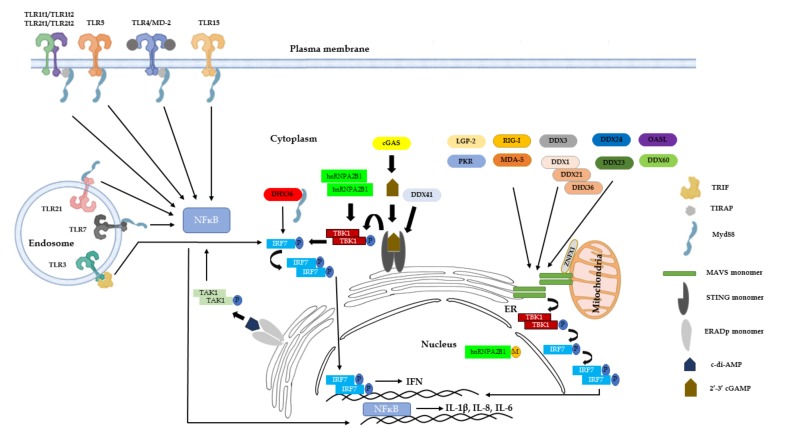
The innate sensing and signaling pathways of chicken, duck and goose. TLR2t1 and TLR2t2 in complex with TLR1t1 or TLR1t2 sense di- and/or triacylated lipopeptides. TLR5 senses bacterial flagellin. The TLR4/MD-2 complex senses bacterial LPS. TLR15 is cleaved and activated by fungal and some bacterial proteases. TLR3, 7, and 21 are endosomal TLR receptors that sense dsRNA, ssRNA, and CpG DNA, respectively. Signaling of all TLRs, except TLR3, is presumed to be via MyD88 adaptor, resulting in the downstream activation of NF-κB and the subsequent production of inflammatory cytokines such as IL-1β, IL-6, and IL-8. TLR3 activation results in TRIF-dependent signaling and downstream activation of interferon regulatory factor 7 (IRF7) and the subsequent production of Type I IFNs. Fully characterized and putative RNA sensors include RIG-I (except chicken), MDA5, LGP2, zinc finger NFX1-type containing 1 (ZNFX1), protein kinase R (PKR), DDX1/DDX21/DHX36 complex, DDX3, DDX23, DDX24, DDX60, and OASL. Fully characterized and putative DNA sensors include cGAS, DDX41, DHX36, and hnRNPA2B1. Cyclic di-nucleotide sensors localized at the endoplasmic reticulum (ER) include STING and ERADp that sense 2′-3′-cGAMP and c-di-AMP, respectively. Figure is partially created through Biorender program. Note: goose carry one isoform each of TLR2 and TLR1.

**Table 1 vetsci-07-00014-t001:** Comparison of the pattern recognition receptors (PRR) and their ligands between human and birds.

Pattern Recognition Receptor (PRR) Class	Human	Chicken	Duck	Goose	Ligand and/or Function
Toll-like receptor (TLR)	TLR2/TLR1	TLR2t1/TLR1t1, TLR2t1/TLR1t2, TLR2t2/TLR1t1	TLR2t1/TLR1t1, TLR2t1/TLR1t2, TLR2t2/TLR1t1	TLR2/TLR1	Tri-acylated lipopetides
	TLR2/TLR6 TLR2t2/TLR16, TLR2t1/TLR1LB	TLR2t2/TLR1t1, TLR2t1/TLR1t2	TLR2t2/TLR1t1, TLR2t1/TLR1t2	TLR2/TLR1	Di-acylated lipopetides
	TLR3	TLR3	TLR3	TLR3	dsRNA
	TLR4/MD-2	TLR4/MD-2	TLR4/MD-2	TLR4/MD-2	LPS
	TLR5	TLR5	TLR5	TLR5	Flagellin
	TLR7	TLR7	TLR7	TLR7	ssRNA
	TLR8	non-functional	non-functional	non-functional	ssRNA
	TLR9	TLR21	TLR21	TLR21	DNA
	Absent	TLR15	TLR15	TLR15	Protease
DEAD/H box helicase	RIG-I	Absent	RIG-I	RIG-I	5’-ppp, short dsRNA
	MDA5	MDA5	MDA5	MDA5	long dsRNA; chMDA5 also senses and responds to short dsRNA
	LGP2	LGP2	LGP2	LGP2	dsRNA and RIG-I/MDA5 regulation
	DDX1/DDX21/DHX36	DDX1/DDX21/DHX36	DDX1/DDX21/DHX36	DDX1/DDX21/DHX36	dsRNA
	DDX3	DDX3	DDX3	DDX3	dsRNA
	DDX23	DDX23	DDX23	DDX23	dsRNA
	DDX24	DDX24	DDX24	DDX24	dsRNA
	DDX60	DDX60	DDX60	DDX60	RIG-I sentinel and RNA degradation via exosome
	DDX41	DDX41	DDX41	DDX41	dsDNA
	DHX36	DHX36	DHX36	DHX36	CpG-A [ODN2216] DNA
	DHX9	Absent	Absent	Absent	CpG-B [ODN2006] DNA
Template independent nucleotidyl transferases	OAS1/2/3	OASL	OASL	OASL	dsRNA
	cGAS	cGAS	cGAS	cGAS	dsDNA
Cyclic di-nucleotide sensor	STING	STING	STING	STING	2’-3’ cGAMP
	ERADp	ERADp	ERADp	ERADp	c-di-AMP
	RECON	Absent	Absent	Absent	c-di-AMP; NFκB negative regulator
Other PRRs	AIM2	Absent	Absent	Absent	dsDNA
	IFI16	Absent	Absent	Absent	dsDNA
	ZBP1/DAI	Absent	Absent	Absent	dsDNA
	HnRNPA2B1	HnRNPA2B1	HnRNPA2B1	HnRNPA2B1	dsDNA

TLR: Toll like receptor, MD-2: Myeloid differentiation protein 2, RIG-I: retinoic acid-inducible gene I, MDA5: melanoma differentiation-associated protein 5, LGP2: Laboratory of genetics and physiology 2, DDX: DEAD box helicase, DHX: DEAH box helicase, OAS: Oligoadenylate synthetase, cGAS: cyclic guanosine-adenosine monophosphate synthase, STING: Stimulator of interferon genes, ERADp: Endoplasmic adaptor protein, RECON: Reductase controlling NFκB, AIM2: Absent in melanoma 2, IFI16: Interferon-inducible protein 16, ZBP1/DAI: Z-DNA binding protein 1/DNA dependent activator of interferon regulatory factors, HnrnpA2B1: Heteronuclear ribonuclear protein A2B1, dsRNA: double-stranded RNA, ssRNA: single-stranded RNA, LPS: lipopolysaccharide, 5’-ppp: 5’ triphosphate moiety of RNA, CpG: Cytosine phosphate guanine, 2’-3’ cGAMP: 2’-3’ linked cyclic guanosine-adenosine monophosphate, AMP: Adenosine monophosphate, NFκB: nuclear factor-κB. For gene accessions, refer Appendix A.

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
