# Peer review of "Avian Pattern Recognition Receptor Sensing and Signaling"

_vetsci, 2020, doi:10.3390/vetsci7010014_

Round 1
Reviewer 1 Report
Very well written and excellent paper. However, one editorial comment that I would make s for the authors to re-write Section 2 to eliminate any major references to mammalian TLR and concentrate only on the poultry TLR pathways (if known).
Author Response
Very well written and excellent paper. However, one editorial comment that I would make s for the authors to re-write Section 2 to eliminate any major references to mammalian TLR and concentrate only on the poultry TLR pathways (if known).
Response: We appreciate constructive suggestions of the reviewer. As per reviewer’s comment, Section 2 was substantially abridged. Sections 2.1.1. and 2.1.2. were removed in the revised manuscript (lines 22-52 of revised manuscript). In addition, a novel DNA sensor hnRNPA2B1 was added in lines 953-971 of revised manuscript.
Reviewer 2 Report
This is review focussed on avian TLR signalling and there are some limitations that require attention.
1) The authors should tell why this review is required. In other words, what is the justification of writing this review. There are reviews on the subject and the authors have not acknowledged them. First pull all the relevant reviews and tell how your review is different from the other.
2) In 2018, there were quite a bit of work done on in ovo delivered TLR ligands in chickens relevant to corona and herpes virus infections and I am not seeing these papers are cited.
3) There are some evidence that, in chickens, TLR4-MyD88 independent pathway exist and these literature can not be ignored.
4) There are some commercial TLR ligands available to use in chickens and those information could be brought.
5) The text is extensive and focussing is required cutting some text.
Author Response
This is review focussed on avian TLR signalling and there are some limitations that require attention.
1.The authors should tell why this review is required. In other words, what is the justification of writing this review. There are reviews on the subject and the authors have not acknowledged them. First pull all the relevant reviews and tell how your review is different from the other.
Response: We appreciate constructive suggestions of the reviewer. The two latest reviews that specifically focused on avian pattern recognition receptors were from 2013[1,2]. These reviews were limited in scope. For instance, review from Keestra et al., focused on chicken TLRs whereas Chen et al., focused on TLRs, NLRs and NLRC5 from chicken, duck and goose, and left many of the details pertaining to duck and goose incomplete or unknown. More importantly, the whole genome sequences of duck and goose were published in 2013 and 2015 respectively [3,4]. Much of the genome sequence of duck and goose has been annotated which led to the identification of PRRs in these species. Furthermore, a plethora of studies already performed functional characterization of duck and goose PRRs which were cited in the present review article. Our submission is the most up to date review on PRRs in chicken, duck and goose. Despite highlighting many PRRs that were characterized in mammalians but not studied in avian (e.g. DDX helicases), the present review highlighted PRRs that are absent in avian or those that evolved in mammalians (e.g. AIM2, IFI16 DNA sensors). There are many reviews that focused on innate immune responses to viral infections such as MDV[5], IBV[6], NDV[7], IAV[8], and ILTV[9] which contained role played by PRRs in the context of these viral infections and a recent review on age-related difference in expression of PRRs in embryos/chicks and adult chickens[10]. However, PRRs that were realized to be critical for certain viruses were not highlighted in these reviews due to recent characterization of these PRRs. For instance, PRR cGAS is critical to restrict DNA viruses such as MDV and possibly ILTV [11,12]. Likewise, recently identified ZNFX1 which acts as a dsRNA sensor in RIG-I/MDA-5-independent manner in mammalians (and also present in avian) may complement RIG-I loss in chicken [13].
As per reviewer’s suggestion, the aim and justification of review were added in the Section 1, lines 53-62 of the revised manuscript.
2. In 2018, there were quite a bit of work done on in ovo delivered TLR ligands in chickens relevant to corona and herpes virus infections and I am not seeing these papers are cited.
Response: As per the reviewer’s comment, studies describing in ovo delivered class B CpG protective effects on MDV, ILTV and IBV challenge were included in the review[14-17]. Lines 982-984 and 991-1007 in the revised manuscript
3. There are some evidence that, in chickens, TLR4-MyD88 independent pathway exist and these literature cannot be ignored.
Response: We thank the reviewer for pointing this out. Study describing TLR4/TRIF-MyD88 independent pathway was cited in the text. Lines 171-173 of revised manuscript
4. There are some commercial TLR ligands available to use in chickens and those information could be brought.
Response: Due to article length restrictions and marketing issues associated with commercial ligands we want to refrain from including such information.
5. The text is extensive and focussing is required cutting some text.
Response: As per the present and former reviewer’s suggestions, section 2 is abridged. Sections 2.1.1. and 2.1.2. were removed in the revised manuscript. Lines 22-52 of revised manuscript. Grammatical errors are corrected in the text
References:
Keestra, A.M.; de Zoete, M.R.; Bouwman, L.I.; Vaezirad, M.M.; van Putten, J.P. Unique features of chicken Toll-like receptors. Dev Comp Immunol 2013, 41, 316-323, doi:10.1016/j.dci.2013.04.009.
Chen, S.; Cheng, A.; Wang, M. Innate sensing of viruses by pattern recognition receptors in birds. Veterinary Research 2013, 44, 82, doi:10.1186/1297-9716-44-82.
Huang, Y.; Li, Y.; Burt, D.W.; Chen, H.; Zhang, Y.; Qian, W.; Kim, H.; Gan, S.; Zhao, Y.; Li, J., et al. The duck genome and transcriptome provide insight into an avian influenza virus reservoir species. Nature Genetics 2013, 45, 776-783, doi:10.1038/ng.2657.
Lu, L.; Chen, Y.; Wang, Z.; Li, X.; Chen, W.; Tao, Z.; Shen, J.; Tian, Y.; Wang, D.; Li, G., et al. The goose genome sequence leads to insights into the evolution of waterfowl and susceptibility to fatty liver. Genome biology 2015, 16, 89, doi:10.1186/s13059-015-0652-y.
Parvizi, P.; Abdul-Careem, M.F.; Haq, K.; Thanthrige-Don, N.; Schat, K.A.; Sharif, S. Immune responses against Marek's disease virus. Anim Health Res Rev 2010, 11, 123-134, doi:10.1017/S1466252310000022.
Chhabra, R.; Chantrey, J.; Ganapathy, K. Immune Responses to Virulent and Vaccine Strains of Infectious Bronchitis Viruses in Chickens. Viral Immunol 2015, 28, 478-488, doi:10.1089/vim.2015.0027.
Kapczynski, D.R.; Afonso, C.L.; Miller, P.J. Immune responses of poultry to Newcastle disease virus. Developmental & Comparative Immunology 2013, 41, 447-453, doi:https://doi.org/10.1016/j.dci.2013.04.012.
Evseev, D.; Magor, K.E. Innate Immune Responses to Avian Influenza Viruses in Ducks and Chickens. Vet Sci 2019, 6, 5, doi:10.3390/vetsci6010005.
Coppo, M.J.C.; Hartley, C.A.; Devlin, J.M. Immune responses to infectious laryngotracheitis virus. Developmental & Comparative Immunology 2013, 41, 454-462, doi:https://doi.org/10.1016/j.dci.2013.03.022.
Alkie, T.N.; Yitbarek, A.; Hodgins, D.C.; Kulkarni, R.R.; Taha-Abdelaziz, K.; Sharif, S. Development of innate immunity in chicken embryos and newly hatched chicks: a disease control perspective. Avian Pathol 2019, 48, 288-310, doi:10.1080/03079457.2019.1607966.
Gao, L.; Li, K.; Zhang, Y.; Liu, Y.; Liu, C.; Zhang, Y.; Gao, Y.; Qi, X.; Cui, H.; Wang, Y., et al. Inhibition of DNA-Sensing Pathway by Marek's Disease Virus VP23 Protein through Suppression of Interferon Regulatory Factor 7 Activation. Journal of virology 2019, 93, e01934-01918, doi:10.1128/jvi.01934-18.
Li, K.; Liu, Y.; Xu, Z.; Zhang, Y.; Luo, D.; Gao, Y.; Qian, Y.; Bao, C.; Liu, C.; Zhang, Y., et al. Avian oncogenic herpesvirus antagonizes the cGAS-STING DNA-sensing pathway to mediate immune evasion. PLoS pathogens 2019, 15, e1007999, doi:10.1371/journal.ppat.1007999.
Wang, Y.; Yuan, S.; Jia, X.; Ge, Y.; Ling, T.; Nie, M.; Lan, X.; Chen, S.; Xu, A. Mitochondria-localised ZNFX1 functions as a dsRNA sensor to initiate antiviral responses through MAVS. Nature cell biology 2019, 21, 1346-1356, doi:10.1038/s41556-019-0416-0.
Abdul-Cader, M.S.; Amarasinghe, A.; Palomino-Tapia, V.; Ahmed-Hassan, H.; Bakhtawar, K.; Nagy, E.; Sharif, S.; Gomis, S.; Abdul-Careem, M.F. In ovo CpG DNA delivery increases innate and adaptive immune cells in respiratory, gastrointestinal and immune systems post-hatch correlating with lower infectious laryngotracheitis virus infection. PLOS ONE 2018, 13, e0193964, doi:10.1371/journal.pone.0193964.
De Silva Senapathi, U.; Abdul-Cader, M.S.; Amarasinghe, A.; van Marle, G.; Czub, M.; Gomis, S.; Abdul-Careem, M.F. The In Ovo Delivery of CpG Oligonucleotides Protects against Infectious Bronchitis with the Recruitment of Immune Cells into the Respiratory Tract of Chickens. Viruses 2018, 10, doi:10.3390/v10110635.
Bavananthasivam, J.; Alkie, T.N.; Astill, J.; Abdul-Careem, M.F.; Wootton, S.K.; Behboudi, S.; Yitbarek, A.; Sharif, S. In ovo administration of Toll-like receptor ligands encapsulated in PLGA nanoparticles impede tumor development in chickens infected with Marek's disease virus. Vaccine 2018, 36, 4070-4076, doi:10.1016/j.vaccine.2018.05.091.
Bavananthasivam, J.; Read, L.; Astill, J.; Yitbarek, A.; Alkie, T.N.; Abdul-Careem, M.F.; Wootton, S.K.; Behboudi, S.; Sharif, S. The effects of in ovo administration of encapsulated Toll-like receptor 21 ligand as an adjuvant with Marek's disease vaccine. Sci Rep 2018, 8, 16370, doi:10.1038/s41598-018-34760-6.
Round 2
Reviewer 2 Report
The authors have addressed the issues in satisfactory manner and I do not have further comments.